# Localized Structured Prediction

**Carlo Ciliberto** [1]
c.ciliberto@imperial.ac.uk

**Francis Bach** [2]
francis.bach@inria.fr

**Alessandro Rudi** [2]
alessandro.rudi@inria.fr

[1] Department of Electrical and Electronic Engineering, Imperial College, London, UK.

[2] INRIA - Département d'informatique, École Normale Supérieure - PSL Research University, Paris, France.

## Abstract

Key to structured prediction is exploiting the problem's structure to simplify the learning process. A major challenge arises when data exhibit a local structure (i.e., are made "by parts") that can be leveraged to better approximate the relation between (parts of) the input and (parts of) the output. Recent literature on signal processing, and in particular computer vision, shows that capturing these aspects is indeed essential to achieve state-of-the-art performance. However, in this context algorithms are typically derived on a case-by-case basis. In this work we propose the first theoretical framework to deal with part-based data from a general perspective and study a novel method within the setting of statistical learning theory. Our analysis is novel in that it explicitly quantifies the benefits of leveraging the part-based structure of a problem on the learning rates of the proposed estimator.

## 1   Introduction

Structured prediction deals with supervised learning problems where the output space is not endowed with a canonical linear metric but has a rich semantic or geometric structure [5, 29]. Typical examples are settings in which the outputs correspond to strings (e.g., captioning [19]), images (e.g., segmentation [1]), rankings [16, 20], points on a manifold [33], probability distributions [24] or protein foldings [18]. While the lack of linearity poses several modeling and computational challenges, this additional complexity comes with a potentially significant advantage: when suitably incorporated within the learning model, knowledge about the structure allows to capture key properties of the data. This could potentially lower the sample complexity of the problem, attaining better generalization performance with less training examples. A natural scenario in this sense is the case where both input and output data are organized into "parts" that can interact with one another according to a specific structure. Examples can be found in computer vision (e.g., segmentation [1], localization [6, 22], pixel-wise classification [41]), speech recognition [4, 40], natural language processing [43], trajectory planing [31] or hierarchical classification [44].

Recent literature on the topic has empirically shown that the local structure in the data can indeed lead to significantly better predictions than global approaches [17, 45]. However in practice, these ideas are typically investigated on a case-by-case basis, leading to ad-hoc algorithms that cannot be easily adapted to new settings. On the theoretical side, few works have considered less specific part-based factorizations [12] and a comprehensive theory analyzing the effect of local interactions between parts within the context of learning theory is still missing.

In this paper, we propose: 1) a novel theoretical framework that can be applied to a wide family of structured prediction settings able to capture potential local structure in the data, and 2) a structured prediction algorithm, based on this framework for which we prove universal consistency and generalization rates. The proposed approach builds on recent results from the structured prediction literature that leverage the concept of *implicit embeddings* [8, 9, 28, 15, 25], also related to [30, 39]. A key contribution of our analysis is to quantify the impact of the part-based structure of the problem on the learning rates of the proposed estimator. In particular, we prove that under natural assumptions on

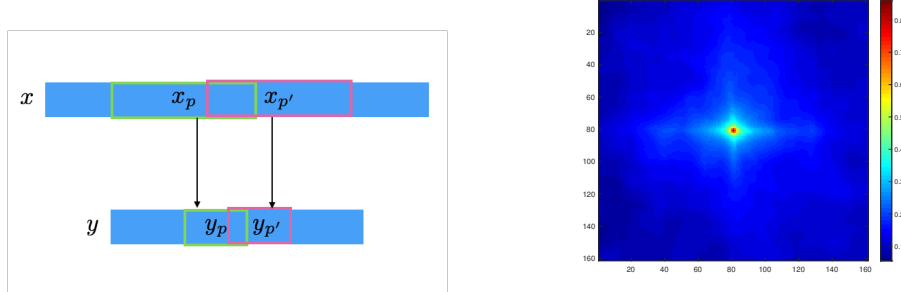

Figure 1: (Left) Between-locality in a sequence-to-sequence setting: each window (part) $y_p$ of the output sequence $y$ is fully determined by the part $x_p$ of the input sequence $x$, for every $p \in P$. (Right) Empirical within-locality $\mathsf{C}_{p,q}$ of 100 images sampled from ImageNet between a $20 \times 20$ patch $q$ and the central patch $p$.

the local behavior of the data, our algorithm benefits *adaptively* from this underlying structure. We support our theoretical findings with experiments on the task of detecting local orientation of ridges in images depicting human fingerprints.

## 2   Learning with Between- & Within-locality

To formalize the concept of locality within a learning problem, in this work we assume that the data is structured in terms of "parts". Practical examples of this setting often arise in image/audio or language processing, where the signal has a natural factorization into patches or sub-sequences. Following these guiding examples, we assume every input $x \in X$ and output $y \in Y$ to be interpretable as a collection of (possibly overlapping) parts, and denote $x_p$ (respectively $y_p$) its $p$-th part, with $p \in P$ a set of part identifiers (e.g., the position and size of a patch in an image). We assume input and output to share same part structure with respect to $P$. To formalize the intuition that the learning problem should interact well with this structure of parts, we introduce two key assumptions: *between-locality* and *within-locality*. They characterize respectively the interplay *between* corresponding input-output parts and the correlation of parts *within* the same input.

**Assumption 1** (Between-locality). *$y_p$ is conditionally independent from $x$, given $x_p$, moreover the probability of $y_p$ given $x_p$ is the same as $y_q$ given $x_q$, for any $p, q \in P$.*

*Between-locality (BL)* assumes that the $p$-th part of the output $y \in Y$ depends only on the $p$-th part of the input $x \in X$, see Fig. 1 (Left) for an intuition in the case of sequence-to-sequence prediction. This is often verified in pixel-wise classification settings, where the class $y_p$ of a pixel $p$ is determined only by the sub-image in the corresponding patch $x_p$. BL essentially corresponds to assuming a joint graphical model on the parts of $x$ and $y$, where each $y_p$ is only connected to $x_p$ but not to other parts.

BL motivates us to focus on a local level by directly learning the relation between input-output parts. This is often an effective strategy in computer vision [22, 45, 17] but intuitively, one that provides significant advantages only when the input parts are not highly correlated with each other: in the extreme case where all parts are identical, there is no advantage in solving the learning problem locally. In this sense it can be useful to measure the amount of "covariance"

$$\mathsf{C}_{p,q} = \mathbb{E}_x \, S(x_p, x_q) - \mathbb{E}_{x,x'} \, S(x_p, x'_q) \tag{1}$$

between two parts $p$ and $q$ of an input $x$, for $S(x_p, x_q)$ a suitable measure of similarity between parts (if $S(x_p, x_q) = x_p x_q$, with $x_p$ and $x_q$ scalars random variables, then $\mathsf{C}_{p,q}$ is the $p, q$-th entry of the covariance matrix of the vector $(x_1, \ldots, x_{|P|})$ ). Here $\mathbb{E}_x S(x_p, x_q)$ and $\mathbb{E}_{x,x'} S(x_p, x'_q)$ measure the similarity between the $p$-th and the $q$-th part of, respectively, the *same* input, and two *independent* ones (in particular $\mathsf{C}_{p,q} = 0$ when the $p$-th and $q$-th part of $x$ are independent). In many applications, it is reasonable to assume that $\mathsf{C}_{p,q}$ decays according to the distance between $p$ and $q$.

**Assumption 2** (Within-locality). *There exists a distance $d : P \times P \to \mathbb{R}$ and $\gamma \geq 0$, such that*

$$|\mathsf{C}_{p,q}| \; \leqslant \; \mathsf{r}^2 \, e^{-\gamma d(p,q)} \qquad with \qquad \mathsf{r}^2 = \sup_{x,x'} |S(x, x')|. \tag{2}$$

*Within-locality (WL)* is always satisfied for $\gamma = 0$. However, when $x_p$ is independent of $x_q$, it holds with $\gamma = \infty$ and $d(p,q) = \delta_{p,q}$ the Dirac's delta. Exponential decays of correlation are typically

observed when the distribution of the parts of $x$ factorizes in a graphical model that connects parts which are close in terms of the distance $d$: although all parts depend on each other, the long-range dependence typically goes to zero exponentially fast in the distance (see, e.g., [26] for mixing properties of Markov chains). Fig. 1 (Right) reports the empirical WL measured on 100 images randomly sampled from ImageNet [13]: each pixel $(i, j)$ reports the value of $\mathsf{C}_{p,q}$ of the central patch $p$ with respect to a $20 \times 20$ patch $q$ centered in $(i, j)$. Here $S(x_p, x_q) = x_p^\top x_q$. We note that $\mathsf{C}_{p,q}$ decreases extremely fast as a function of the distance $\|p - q\|$, suggesting that Assumption 2 holds for a large value of $\gamma$.

**Contributions.** In this work we present a novel structured prediction algorithm that adaptively leverages locality in the learning problem, when present (Sec. 4). We study the generalization properties of the proposed estimator (Sec. 5), showing that it is equivalent to the state of the art in the worst case scenario. More importantly, if the locality Assumptions 1 and 2 are satisfied, we prove that our learning rates improve proportionally to the number $|P|$ of parts in the problem. Here we give an informal version of this main result, reported in more detail in Thm. 4 (Sec. 5). Below we denote by $\widehat{f}$ the proposed estimator, by $\mathcal{E}(f)$ the expected risk of a function $f : X \to Y$ and $f^* = \operatorname{argmin}_f \mathcal{E}(f)$.

**Theorem 1** (Informal - Learning Rates & Locality). *Under mild assumptions on the loss and the data distribution, if the learning problem is local (Assumptions 1 and 2), there exists $c_0 > 0$ such that*

$$\mathbb{E}\left[\mathcal{E}(\widehat{f}) - \mathcal{E}(f^*)\right] \leqslant c_0 \left(\frac{\mathsf{s}}{n|P|}\right)^{1/4}, \qquad \mathsf{s} = \frac{\mathsf{r}^2}{|P|} \sum_{p,q=1}^{|P|} e^{-\gamma d(p,q)}, \qquad (3)$$

*where the expectation is taken with respect to the sample of $n$ input-output points used to train $\widehat{f}$.*

In the worst-case scenario $\gamma = 0$ (no exponential decay of the covariance between parts), the bound in (3) scales as $1/n^{1/4}$ (since $\mathsf{s} = \mathsf{r}^2 |P|$) recovering [8], where no structure is assumed on the parts. However, as soon as $\gamma > 0$, $\mathsf{s}$ can be upper bounded by a constant independent of $|P|$ and thus the rate scales as $1/(|P|n)^{1/4}$, accelerating proportionally to the number of parts. In this sense, Thm. 1 shows the significant benefit of making use of locality. The following example focuses on the special case of sequence-to-sequence prediction.

**Example 1** (Locality on Sequences). *As depicted in Fig. 1, for discrete sequences we can consider parts (e.g., windows) indexed by $P = \{1, \dots, |P|\}$, with $d(p, q) = |p - q|$ for $p, q \in P$ (see Appendix K.1 for more details). In this case, Assumption 2 leads to*

$$\mathsf{s} \leqslant 2\mathsf{r}^2(1 - e^{-\gamma})^{-1}, \qquad (4)$$

*which for $\gamma > 0$ is bounded by a constant not depending on the number of parts. Hence, Thm. 1 guarantees a learning rate of order $1/(n|P|)^{1/4}$, which is significanlty faster than the rate $1/n^{1/4}$ of methods that do not leverage locality such as [8]. See Sec. 6 for empirical support to this observation.*

## 3 Problem Formulation

We denote by $X, Y$ and $Z$ respectively the *input space*, *label space* and *output space* of a learning problem. Let $\rho$ be a probability measure on $X \times Y$ and $\triangle : Z \times Y \times X \to \mathbb{R}$ a loss measuring prediction errors between a label $y \in Y$ and a output $z \in Z$, possibly parametrized by an input $x \in X$. To stress this interpretation we adopt the notation $\triangle(z, y|x)$. Given a finite number of $(x_i, y_i)_{i=1}^n$ independently sampled from $\rho$, our goal is to approximate the minimizer $f^*$ of the *expected risk*

$$\min_{f:X \to Z} \mathcal{E}(f), \quad \text{with} \quad \mathcal{E}(f) = \int \triangle(f(x), y|x) \, d\rho(x, y). \qquad (5)$$

**Loss Made by Parts.** We formalize the intuition introduced in Sec. 2 that data are decomposable into parts: we denote the sets of *parts* of $X, Y$ and $Z$ by respectively $[X], [Y]$ and $[Z]$. These are abstract sets that depend on the problem at hand (see examples below). We assume $P$ to be a set of part "indices" equipped with a selection operator $X \times P \to [X]$ denoted $(x, p) \mapsto [x]_p$ (analogously for $Y$ and $Z$). When clear from context, we will use the shorthand $x_p = [x]_p$. For simplicity, in the following we will assume $P$ be finite, however our analysis generalizes also to the infinite case (see

supplementary material). Let $\pi(\cdot|x)$ be a probability distribution over the set of parts $P$, conditioned with respect to an input $x \in X$. We study loss functions $\triangle$ that can be represented as

$$\triangle(z,y|x) = \sum_{p \in P} \pi(p|x) \, L_p(z_p, y_p|\, x_p). \tag{6}$$

The collection of $(L_p)_{p \in P}$ is a family of loss functions $L_p : [Z] \times [Y] \times [X] \to \mathbb{R}$, each comparing the $p$-th part of a label $y$ and output $z$. For instance, in an image processing scenario, $L_p$ could measure the similarity between the two images at different locations and scales, indexed by $p$. In this sense, the distribution $\pi(p|x)$ allows to weigh each $L_p$ differently depending on the application (e.g., mistakes at large scales could be more relevant than at lower scales). Various examples of parts and concrete cases are illustrated in the supplementary material, here we report an extract.

**Example 2** (Sequence to Sequence Prediction). *Let $X = A^k$, $Y = Z = B^k$ for two sets $A, B$ and $k \in \mathbb{N}$ a fixed length. We consider in this example parts that are windows of length $l \leqslant k$. Then $P = \{1, \ldots, k - l + 1\}$ where $p \in P$ indexes the window $x_p = (x^{(p)}, \ldots, x^{(p+l-1)})$, with $x \in X$, where we have denoted $x^{(s)}$ the $s$-th entry of the sequence $x \in X$, analogous definition for $y_p, z_p$. Finally, we choose the loss $L_p$ to be the 0-1 distance between two strings of same length $L_p(z_p, y_p|x) = \mathbf{1}(z_p \neq y_p)$. Finally, we can choose $\pi(p|x) = 1/|P|$, leading to a loss function $\triangle(z,y|x) = \frac{1}{|P|} \sum_{p \in P} \mathbf{1}(z_p \neq y_p)$, which is common in the context of Conditional Random Fields (CRFs) [21].*

The example above, highlights a tight connection between the framework considerd in this work and the literature of CRFs. However, we care to stress that the two approaches differ by the way they interpret the concepts of loss (used to evaluate fitting errors at training time) and the score functions (used to estimate predictions at inference time). Specifically, while such functions are two separate entities in CRF settings, they essentially coincide in our framework (i.e. the score is a linear combination of loss functions). However, as shown in Example 2, the resulting score functions for both CRFs and our approach have essentially the same structure. Hence they ultimately lead to the same inference problem [40]. We conclude this section by providing additional examples of loss functons decomposable into parts.

**Remark 1** (Examples of Loss Functions by Parts). *Several loss functions used in machine learning have a natural formulation in terms of (6). Notable examples are the Hamming distance [10, 42, 11], used in settings such as hierarchical classification [44], computer vision [29, 45, 41] or trajectory planning [31] to name a few. Also, loss functions used in natural language processing, such as the precision/recall and $F1$ score can be written in this form. Finally, we point out that multi-task learning settings [27] can be seen as problem by parts, with the loss corresponding to the sum of standard regression/classification loss functions (least-squares, logistic, etc.) over the tasks/parts.*

## 4 Algorithm

In this section we introduce our estimator for structured prediction problems with parts. Our approach starts with an auxiliary step for dataset generation that explicitly extracts the parts from the data.

**Auxiliary Dataset Generation.** The locality assumptions introduced in Sec. 2 motivate us to learn the local relations between individual parts $p \in P$ of each input-output pair. In this sense, given a training dataset $\mathcal{D} = (x_i, y_i)_{i=1}^n$ a first step would be to extract a new, part-based dataset $\{(x_p, p, y_p) \,|\, (x, y) \in \mathcal{D}, \ p \in P\}$. However in most applications the cardinality $|P|$ of the set of parts can be very large (possibly infinite as we discuss in the Appendix) making this process impractical. Instead, we generate an *auxiliary dataset* by randomly sub-sampling $m \in \mathbb{N}$ elements from the part-based dataset. Concretely, for $j \in \{1, \ldots, m\}$, we first sample $i_j$ according to the uniform distribution $U_n$ on $\{1, \ldots, n\}$, set $\chi_j = x_{i_j}$, sample $p_j \sim \pi(\cdot \mid \chi_j)$ and finally set $\eta_j = [y_{i_j}]_{p_j}$. This leads to the auxiliary dataset $\mathcal{D}' = (\chi_j, p_j, \eta_j)_{j=1}^m$, as summarized in the GENERATE routine of Alg. 1.

**Estimator.** Given the auxiliary dataset, we propose the estimator $\widehat{f} : X \to Z$, such that $\forall x \in X$

$$\widehat{f}(x) = \operatorname*{argmin}_{z \in Z} \sum_{p \in P} \sum_{j=1}^m \alpha_j(x,p) \left[ \pi(p|x) \, L_p(z_p, \eta_j|x_p) \right]. \tag{7}$$

The functions $\alpha_j : X \times P \to \mathbb{R}$ are *learned* from the auxiliary dataset and are the fundamental components allowing our estimator to capture the part-based structure of the learning problem. Indeed,

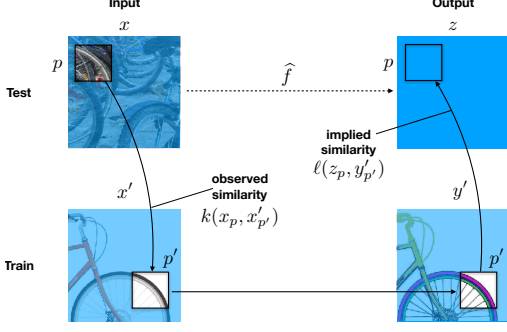

**Input**
$x$

**Output**
$z$

Test

$\widehat{f}$

implied similarity
$\ell(z_p, y'_{p'})$

observed similarity
$k(x_p, x'_{p'})$

$x'$

$y'$

Train

Figure 2: Illustration of the prediction process for the Localized Structured Prediction Estimator (7) for a hypothetical computer vision application.

---

**Algorithm 1**

---

**Input:** training set $(x_i, y_i)_{i=1}^n$, distributions $\pi(\cdot|x)$ a reproducing kernel $k$ on $X \times P$, hyper-parameter $\lambda > 0$, auxiliary dataset size $m \in \mathbb{N}$.

GENERATE the auxiliary set $(\eta_j, \chi_j, p_j)_{j=1}^m$:
  Sample $i_j \in U_n(\cdot)$. Set $\chi_j = x_{i_j}$.
  Sample $p_j \sim \pi(\cdot|\chi_j)$. Set $\eta_j = [y_{i_j}]_{p_j}$.

LEARN the coefficients for the map $\alpha$:
  Set $\mathbf{K}$ with $\mathbf{K}_{jj'} = k((\chi_j, p_j), (\chi_{j'}, p_{j'}))$.
  $\mathbf{A} = (\mathbf{K} + m\lambda I)^{-1}$.

**Return** the map $\alpha : (x, p) \mapsto \mathbf{A}\, v(x, p) \in \mathbb{R}^m$
  with $v(x, p)_j = k((\chi_j, p_j), (x, p))$.

---

for any test point $x \in X$ and part $p \in P$, the value $\alpha_j(x, p)$ can be interpreted as a measure of how similar $x_p$ is to the $p_j$-th part of the auxiliary training point $\chi_j$. For instance, assume $\alpha_j(x, p)$ to be an approximation of the delta function that is 1 when $x_p = [\chi_j]_{p_j}$ and 0 otherwise. Then,

$$\alpha_j(x, p)\, L_p(z_p, \eta_j | x_p) \quad \approx \quad \delta(x_p, [\chi_j]_{p_j})\, L_p(z_p, \eta_j | x_p), \tag{8}$$

which implies essentially that

$$x_p \approx [\chi_j]_{p_j} \implies z_p \approx \eta_j. \tag{9}$$

In other words, if the $p$-th part of test input $x$ and the $p_j$-th part of the auxiliary training input $\chi_j$ (i.e., the $p_j$-th part of the training input $x_{i_j}$) are deemed similar, then the estimator will encourage the $p$-th part of the test output $z$ to be similar to the auxiliary part $\eta_j$. This process is illustrated in Fig. 2 for an ideal computer vision application: for a given test image $x$, the $\alpha$ scores detect a similarity between the $p$-th patch of $x$ and the $p_j$-th patch of the training input $x_{i_j}$. Hence, the estimator will enforce the $p$-th patch of the output $z$ to be similar to the $p_j$-th patch of the training label $y_{i_j}$.

**Learning $\alpha$.** In line with previous work on structured prediction [8], we learn each $\alpha_j$ by solving a linear system for a problem akin to kernel ridge regression (see Sec. 5 for the theoretical motivation). In particular, let $k : (X \times P) \times (X \times P) \to \mathbb{R}$ be a positive definite kernel, we define

$$(\alpha_1(x, p), \ldots, \alpha_m(x, p))^\top = (\mathbf{K} + m\lambda I)^{-1} v(x, p), \tag{10}$$

where $\mathbf{K} \in \mathbb{R}^{m \times m}$ is the empircal kernel matrix with entries $\mathbf{K}_{jh} = k((\chi_j, p_j), (\chi_h, p_h))$ and $v(x, p) \in \mathbb{R}^m$ is the vector with entries $v(x, p)_j = k((\chi_j, p_j), (x, p))$. Training the proposed algorithm, consists in precomputing $\mathbf{A} = (\mathbf{K} + m\lambda I)^{-1}$ to evaluate the coefficients $\alpha$ as detailed by the LEARN routine in Alg. 1. While computing $\mathbf{A}$ amounts to solving a linear system, which requires $O(m^3)$ operations, we note that it is possible to achieve the same statistical accuracy with reduced complexity $O(m\sqrt{m})$ by means of low rank approximations (see [14, 32]).

**Remark 2 (Evaluating $\widehat{f}$).** *According to (7), evaluating $\widehat{f}$ on a test point $x \in X$ consists in solving an optimization problem over the output space $Z$. This is a standard strategy in structured prediction, where an optimization protocol is derived on a case-by-case basis depending on both $\triangle$ and $Z$ (see, e.g., [29]). Hence, from a computational viewpoint, the inference step in this work is not more demanding than previous methods (while also enjoying strong theoretical guarantees on the prediction perfomance, as discussed in Sec. 5). Moreover, the specific form of our estimator suggests a general stochastic meta-algorithm to address the inference problem in special settings. In particular, we can reformulate (7) as*

$$\widehat{f}(x) = \underset{z \in Z}{\mathrm{argmin}}\; \mathbb{E}_{j,p}\, h_{j,p}(z|x), \tag{11}$$

*with $p$ sampled according to $\pi$, $j \in \{1, \ldots, m\}$ sampled according to the weights $\alpha_j$ and $h_{j,p}$ suitably defined in terms of $L_p$. When the $h_{j,p}$ are (sub)differentiable, (11) can be effectively addressed by stochastic gradient methods (SGM). In Alg. 3 in Appendix J we give an example of this strategy.*

## 5 Generalization Properties of Structured Prediction with Parts

In this section we study the statistical properties for the proposed algorithm, with particular attention to the impact of locality on learning rates, see Thm. 4 (for a complete analysis of univeral consistency and learning rates without locality assumptions, see Appendices F and H). Our analysis leverages the assumption that the loss function $\triangle$ is a *Structure Encoding Loss Function (SELF) by Parts*.

**Definition 1** (SELF by Parts). *A function* $\triangle : Z \times Y \times X \to \mathbb{R}$ *is a* Structure Encoding Loss Function (SELF) by Parts *if it admits a factorization in the form of* (6) *with functions* $L_p : [Z] \times [Y] \times [X] \to \mathbb{R}$, *and there exists a separable Hilbert space* $\mathcal{H}$ *and two bounded maps* $\psi : [Z] \times [X] \times P \to \mathcal{H}$, $\varphi : [Y] \to \mathcal{H}$ *such that for any* $\zeta \in [Z]$, $\eta \in [Y]$, $\xi \in [X]$, $p \in P$

$$L_p(\zeta, \eta | \xi) = \langle \psi(\zeta, \xi, p), \varphi(\eta) \rangle_{\mathcal{H}}. \tag{12}$$

The definition of "SELF by Parts" specializes the definition of SELF in [9] and in the following we will always assume $\triangle$ to satisfy it. Indeed, Def. 1 is satisfied when the spaces of parts involved are discrete sets and it is rather mild in the general case (see [8] for an exhaustive list of examples). Note that when $\triangle$ is SELF, the solution of (5) is completely characterized in terms of the conditional expectation (related to the *conditional mean embedding* [7, 23, 36, 34]) of $\varphi(y_p)$ given $x$, denoted by $g^* : X \times P \to \mathcal{H}$, as follows.

**Lemma 2.** *Let* $\triangle$ *be SELF and* $Z$ *compact. Then, the minimizer of* (5) *is* $\rho_X$-*a.e. characterized by*

$$f^*(x) = \underset{z \in Z}{\operatorname{argmin}} \sum_{p \in P} \pi(p|x) \langle \psi(z_p, x_p, p), g^*(x, p) \rangle_{\mathcal{H}}, \qquad g^*(x, p) = \int_Y \varphi(y_p) d\rho(y|x). \tag{13}$$

Lemma 2 (proved in Appendix C) shows that $f^*$ is completely characterized in terms of the conditional expectation $g^*$, which indeed plays a key role in controlling the learning rates of $\widehat{f}$. In particular, we investigate the learning rates in light of the two assumptions of between- and within-locality introduced in Sec. 2. To this end, we first study the direct effects of these two assumptions on the learning framework introduced in this work.

**The effect of Between-locality.** We start by observing that the between-locality between parts of the inputs and parts of the output allows for a refined characterization of the conditional mean $g^*$.

**Lemma 3.** *Let* $g^*$ *be defined as in* (13). *Under* Assumption 1, *there exists* $\bar{g}^* : [X] \to \mathcal{H}$ *such that*

$$g^*(x, p) = \bar{g}^*(x_p) \qquad \forall x \in X, \ p \in P. \tag{14}$$

Lemma 3 above shows that we can learn $g^*$ by focusing on a "simpler" problem, identified by the function $\bar{g}^*$ acting only the parts $[X]$ of $X$ rather than on the whole input directly (for a proof see Lemma 21 in Appendix G). This motivates the adoption of the restriction kernel [6], namely a function $k : (X \times P) \times (X \times P) \to \mathbb{R}$ such that

$$k((x, p), (x', q)) = \bar{k}(x_p, x_q), \tag{15}$$

which, for any pair of inputs $x, x' \in X$ and parts $p, q \in P$, measures the similarity between the $p$-part of $x$ and the $q$-th part of $q$ via a kernel $\bar{k} : [X] \times [X] \to \mathbb{R}$ on the parts of $X$. The restriction kernel is a well-established tool in structured prediction settings [6] and it has been observed to be remarkably effective in computer vision applications [22, 45, 17].

**The effect of Within-locality.** We recall that within-locality characterizes the statistical correlation between two different parts of the input (see Assumption 2). To this end we consider the simplified scenario where the parts are sampled from the uniform distribution on $P$, i.e., $\pi(p|x) = \frac{1}{|P|}$ for any $x \in X$ and $p \in P$. While more general situations can be considered, this setting is useful to illustrate the effect we are interested in this work. We now define some important quantities that characterize the learning rates under locality,

$$\mathsf{C}_{p,q} = \mathbb{E}_{x,x'} \left[ \bar{k}(x_p, x_q)^2 - \bar{k}(x_p, x'_q)^2 \right], \qquad \mathsf{r} = \sup_{x \in X, p \in P} \bar{k}(x_p, x_p). \tag{16}$$

It is clear that the terms $\mathsf{C}_{p,q}$ and $\mathsf{r}$ above correspond respectively to the correlations introduced in (1) and the scale parameter introduced in (2), with similarity function $S = \bar{k}^2$. Let $\widehat{f}$ be the structured prediction estimator in (7) learned using the restriction kernel in (15) based on $\bar{k}$ and denote by $\bar{\mathcal{G}}$ the

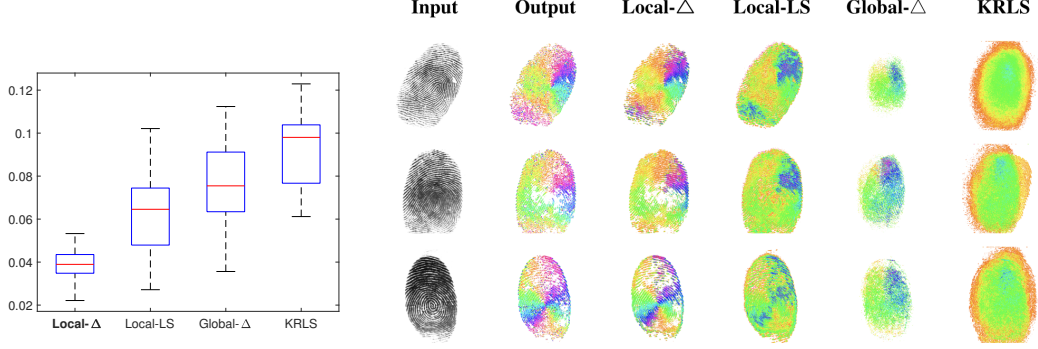

Figure 3: Learning the direction of ridges in fingerprint images. (Left) Examples of ground truths and predictions with pixels' color corresponding to the local direction of ridges. (Right) Test error according to $\triangle$ in (18).

space of functions $\bar{\mathcal{G}} = \mathcal{H} \otimes \bar{\mathcal{F}}$ with $\bar{\mathcal{F}}$ the reproducing kernel Hilbert space [3] associated to $\bar{k}$. In particular, in the following we will consider the standard assumption in the context of non-parametric estimation [7] on the regularity of the target function, which in our context reads as $\bar{g}^* \in \bar{\mathcal{G}}$. Finally we introduce $\mathsf{c}_\triangle^2 = \sup_{z \in Z, x \in X} \frac{1}{|P|} \sum_{p \in P} \|\psi(z, x, p)\|_{\mathcal{H}}^2$ to measure the "complexity" of the loss $\triangle$ w.r.t. the representation induced by SELF decomposition (Def. 1) analogously to Thm. 2 of [8].

**Theorem 4** (Learning Rates & Locality). *Under Assumptions 1 and 2 with $S = \bar{k}^2$, let $\bar{g}^*$ satisfying Lemma 3, with $\bar{\mathsf{g}} = \|\bar{g}^*\|_{\bar{\mathcal{G}}} < \infty$. Let $\mathsf{s}$ be as in (3). When $\lambda = (\mathsf{r}^2/m + \mathsf{s}/(|P|n))^{1/2}$, then*

$$\mathbb{E} \, \mathcal{E}(\widehat{f}) - \mathcal{E}(f^*) \; \leqslant \; 12 \, \mathsf{c}_\triangle \, \bar{\mathsf{g}} \left( \frac{\mathsf{r}^2}{m} + \frac{\mathsf{r}^2}{|P|n} + \frac{\mathsf{s}}{|P|n} \right)^{1/4} . \tag{17}$$

The proof of the result above can be found in Appendix G.1. We can see that between- and within-locality allow to refine (and potentially improve) the bound of $n^{-1/4}$ from structured prediction without locality [8] (see also Thm. 5 in Appendix F). In particular, we observe that the adoption of the restriction kernel in Thm. 4 allows the structured prediction estimator to leverage the within-locality, gaining a benefit proportional to the magnitude of the parameter $\gamma$. Indeed $\mathsf{r}^2 \leqslant \mathsf{s} \leqslant \mathsf{r}^2|P|$ by definition. More precisely, if $\gamma = 0$ (e.g., all parts are identical copies) then $\mathsf{s} = \mathsf{r}^2|P|$ and we recover the rate of $O(n^{-1/4})$ of [8], while if $\gamma$ is large (the parts are almost not correlated) then $\mathsf{s} = \mathsf{r}^2$ and we can take $m \propto n|P|$ achieving a rate of the order of $O\big((n|P|)^{-1/4}\big)$. We clearly see that depending on the amount of within-locality in the learning problem, the proposed estimator is able to gain significantly in terms of finite sample bounds.

## 6 Empirical Evaluation

We evaluate the proposed estimator on simulated as well as real data. We highlight how locality leads to improved generalization performance, in particular when only few training examples are available.

**Learning the Direction of Ridges for Fingerprint.** Similarly to [37], we considered the problem of detecting the pointwise direction of ridges in a fingerprint image on the FVC04 dataset[1] comprising 80 grayscale $640 \times 480$ input images depicting fingerprints and corresponding output images encoding in each pixel the local direction of the ridges of the input fingerprint as an angle $\theta \in [-\pi, \pi]$. A natural loss function is the average pixel-wise error $\sin(\theta - \theta')^2$ between a ground-truth angle $\theta$ and the predicted $\theta'$ according to the geodesic distance on the sphere. To apply the proposed algorithm, we consider the following representation of the loss in term of parts: let $P$ be the collection of patches of dimension $20 \times 20$ and equispaced each $5 \times 5$ pixels[2] so that each pixel belongs exactly to 16 patches. For all $z, y \in \mathbb{R}^{640 \times 480}$, the average pixel-wise error is

$$\triangle(z, y) = \frac{16}{|P|} \sum_{p \in P} L(z_p, y_p), \qquad \text{with} \qquad L(\zeta, \eta) = \frac{1}{20 \times 20} \sum_{i,j=1}^{20} \sin([\zeta]_{ij}, [\eta]_{ij})^2, \tag{18}$$

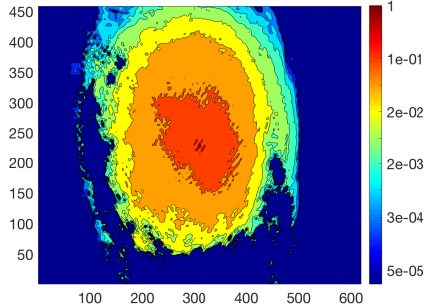

Figure 4: Empirical estimation of *within-locality* for the central patch of the fingerprints dataset.

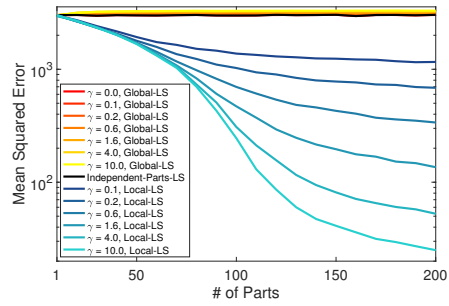

Figure 5: Effect of within-locality w.r.t. $\gamma$ and $|P|$: *Global-LS* vs. *IndependentParts-LS* vs. *Local-LS* (ours).

where $\zeta = z_p, \eta = y_p \in [-\pi, \pi]^{20 \times 20}$ are the extracted patches and $[\cdot]_{ij}$ their value at pixel $(i, j)$.

We compared our approach using $\triangle$ (*Local-$\triangle$*) or least-squares (*Local-LS*) with competitors that do not take into account the local structure of the problem, namely standard vector-valued kernel ridge regression (*KRLS*) [7] and the structured prediction algorithm in [8] with $\triangle$ loss ($\triangle$-*Global*). We used a Gaussian kernel on the input (for the local estimators the restriction kernel in (15) with $\bar{k}$ Gaussian). We randomly sampled $50/30$ images for training/testing, performing 5-fold cross-validation on $\lambda$ in $[10^{-6}, 10]$ (log spaced) and the kernel bandwidth in $[10^{-3}, 1]$. For Local-$\triangle$ and Local-LS we built an auxiliary set with $m = 30000$ random patches (see Sec. 4), sampled from the 50 training images.

**Results**. Fig. 3 (Left) reports the average prediction error across 10 random train-test splits. We make two observations: first, methods that leverage the locality in the data are consistently superior to their "global" counterparts, supporting our theoretical results in Sec. 5 that the proposed estimator can lead to significantly better performance, in particular when few training points are available. Second, the experiment suggests that choosing the right loss is critical, since exploiting locality without the right loss (i.e., Local-LS in the figure) generally leads to worse performance. The three sample predictions in Fig. 3 (Right) provide more qualitative insights on the models tested. In particular while both locality-aware methods are able to recover the correct structure of the fingerprints, only combining this information with the loss $\triangle$ leads to accurate recovery of the ridge orientation.

**Within-locality**. In Fig. 4 we visualize the (empirical) within-locality of the central patch $p$ for the fingerprint dataset. The figure depicts $\mathsf{C}_{p,q}$ (defined in (16)) for $q \in P$, with the $(i, j)$-th pixel in the image corresponding to $\mathsf{C}_{p,q}$ with $q$ the $20 \times 20$ patch centered in $(i, j)$. The fast decay of these values as the distance from the central patch $p$ increase, suggests that within-locality holds for a large value of $\gamma$, possibly justifying the good performance exhibited by (*Local-$\triangle$*) in light of Thm. 4.

**Simulation: Within-Locality.** We complement our analysis with synthetic experiments where we control the "amount" of within-locality $\gamma$. We considered a setting where input points are vectors $x \in \mathbb{R}^{k|P|}$ comprising $|P|$ parts of dimension $k = 1000$. Inputs are sampled according to a normal distribution with zero mean and covariance $\Sigma(\gamma) = M(\gamma) \otimes I$, where $M(\gamma) \in \mathbb{R}^{|P| \times |P|}$ has entries $M(\gamma)_{pq} = e^{-\gamma d(p,q)}$ and $d(p, q) = |p - q|/|P|$. By design, as $\gamma$ grows $\mathsf{C}$ varies from being rank-one (all parts are identical copies) to diagonal (all parts are independently sampled).

To isolate the effect of within-locality on learning, we tested our estimator on a linear multitask (actually vector-valued) regression problem with least-squares loss $\triangle$. We generated datasets $(x_i, y_i)_{i=1}^n$ of size $n = 100$ for training and $n = 1000$ for testing, with $x_i$ sampled as described above and $y_i = w^\top x_i + \epsilon$ with noise $\epsilon \in \mathbb{R}^{k|P|}$ sampled from an isotropic Gaussian with standard deviation $0.5$. To guarantee *between*-locality to hold, we generated the target vector $w = [\bar{w}, \ldots, \bar{w}] \in \mathbb{R}^{k|P|}$ by concatenating copies of a $\bar{w} \in \mathbb{R}^k$ sampled uniformly on the radius-one ball. We performed regression with linear restriction kernel on the parts/subvectors (*Local-LS*) on the "full" auxiliary dataset $([x_i]_p, [y_i]_p)$ with $1 \leqslant i \leqslant n$ and $1 \leqslant p \leqslant |P|$, and compared it with standard linear regression (*Global-LS*) on the original dataset $(x_i, y_i)_{i=1}^n$ and linear regression performed independently for each (local) subdataset $([x_i]_p, [y_i]_p)_{i=1}^n$ (*IndependentParts - LS*). The parameter $\lambda$ was chosen by hold-out cross-validation in $[10^{-6}, 10]$ (log spaced).

Fig. 5 reports the (log scale) mean square error (MSE) across 100 runs of the two estimators for increasing values of $\gamma$ and $|P|$. In line with Thm. 4, when $\gamma$ and $|P|$ are large, Local-LS significantly outperforms both $i$) Global-LS, which solves one single problem jointly and does not benefit within-locality, and $ii$) IndependentParts-LS, which is insensitive to the between-locality across parts and solves each local prediction problem in isolation. For a smaller $\gamma$, such advantage becomes less prominent even when the number of parts is large. This is expected since for $\gamma = 0$ the input parts are extremely correlated and there is no within locality that can be exploited.

## 7 Conclusion

We proposed a novel approach for structured prediction in presence of locality in the data. Our method builds on [8] by incorporating knowledge of the parts directly within the learning model. We proved the benefits of locality by showing that, under a low-correlation assumption on the parts of the input (within locality), the learning rates of our estimator can improve proportionally to the number of parts in the data. To obtain this result we additionally introduced a natural assumption on the conditional independence between input-output parts (between locality), which provides also a formal justification for adoption of the so-called "restriction kernel", previously proposed in the literature, as a mean to lower the sample complexity of the problem. Empirical evaluation on synthetic as well as real data shows that our approach offers significant advantages when few training points are available and leveraging structural information such as locality is crucial to achieve good prediction performance. We identify two main directions for future work: 1) consider settings where the parts are unknown (or "latent") and need to be discovered/learned from data; 2) Consider more general locality assumptions. In particular, we argue that Assumption 2 (WL) might be weakened to account for different (but related) local input-output relations across adjacent parts.

## Footnotes

[1] http://bias.csr.unibo.it/fvc2004, DB1_B. The output is obtained by applying $7 \times 7$ Sobel filtering.

[2] For simplicity we assume "circular images", namely $[x]_{i,j} = [x]_{(i \mod 640),(j \mod 480)}$.

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
