[Supplementary Material · localized-all-final.pdf]

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

# Supplementary Material of:
# Localized Structured Prediction

In this appendix we provide further background to the main discussion and results in the main sections of the current work. In particular:

- Appendix A introduces a generalization of the proposed framework to account for a larger family of structured prediction problems where locality can be exploited.
- Appendix B introduces the notation and auxiliary results that will be useful to prove the results discussed in this work.
- Appendix C discusses the derivation of the structured prediction estimator proposed and studied in this work.
- Appendix D extends the Comparison inequality for the SELF estimator in [8] to the case where the locality of the problem can be exploited.
- Appendix E provides an analytical decomposition of a bound for the excess risk of the proposed estimator that is then used to prove the learning rates of the proposed estimator without and with parts (respectively Appendices F and G) and also the universal consistency (Appendix H).
- Appendix I compares the proposed framework with structured prediction (without parts) in [8].
- Appendix J provides more details on the problem of learning and evaluating the estimator proposed in this work.
- Appendix K discusses in more detail loss functions considered in the literature that can be decomposed into "parts".

**An overview of the main result in Thm. 4.**

For the sake of clarity, before delving in the discussion below, we discuss here how the main result of this work, namely Thm. 4, is situated within the appendix. While the formal proof is given in Appendix G.1, here we highlight and reference the key results used to this purpose. The main analysis in this sense can be found in Appendices E to G.

In particular, the proof hinges on three main components: first, we study the conditional expectation $g^*$ introduced in Lemma 2 in terms of an estimator $\widehat{g}$. In Appendix C we prove that this estimator is tightly connected to our structured prediction estimator in (7) according to the comparison inequality

$$\mathcal{E}(\widehat{f}) - \mathcal{E}(f^*) \leqslant \mathsf{c}_\triangle \|\widehat{g} - g^*\|_{L^2(X \times P, \pi\rho_X, \mathcal{H})}. \tag{19}$$

proved in Thm. 9 in Appendix C.

Second, we study the quantity $\|\widehat{g} - g^*\|_{L^2(X \times P, \pi\rho_X, \mathcal{H})}$, providing an analytic decomposition of this error in Thm. 11 in Appendix E.

Finally, in Appendix F we consider how each term in such analytical decomposition can be controlled in expectation with respect to a training dataset randomly sampled from the underlying distribution $\rho$. Putting together all these independent bounds we are able to characterize the excess risk bounds for our estimator $\widehat{f}$ in the *general setting where locality does not necessary hold*, which is reported below and proved at the end of Appendix F.3.

**Theorem 5.** *Let $\widehat{f}$ as in (7) with i.i.d. training set and auxiliary dataset sampled according to Alg. 1. Let $\triangle$ be SELF, $Z$ compact, $g^* \in \mathcal{G}$ and $\lambda \geq (\mathsf{r}^2/m + \mathsf{q}/n)^{1/2}$. Then*

$$\mathbb{E}\left[\mathcal{E}(\widehat{f}) - \mathcal{E}(f^*)\right] \leqslant 12\,\mathsf{c}_\triangle\,\mathsf{g}\left(\frac{\mathsf{r}^2}{\lambda m} + \frac{\mathsf{q}}{\lambda n} + \lambda\right)^{1/2}. \tag{20}$$

Here we have introduced the quantity

$$\mathsf{q} = \mathbb{E}_{x,x'}\mathbb{E}_{p,q|x,r|x'}\,\mathsf{C}_{p,q}(x,x') \qquad \mathsf{C}_{p,q}(x,x') = \left[k((x,p),(x,q))^2 - k((x,p),(x',r))^2\right], \tag{21}$$

where $\mathbb{E}_{p,q|x}[\cdot]$ is a shorthand for $\sum_{p,q \in P} \pi(p|x)\pi(q|x)[\cdot]$ (analogously for $\mathbb{E}_{r|x}$). It can be seen that this quantity allows to capture and leverage the within-locality assumption. In particular, it will allow us to quantify explicitly the advantages of using our locality-aware estimator.

The result above explicitly shows how the quantities measuring the within locality do affect the constants in the learning rates of the proposed estimator. By combining Thm. 5 with Assumptions 1 and 2 and leveragin the locality properties of the restriction kernel introduced in (15), we are then able to prove Thm. 4 as desired. As mentioned, the details of this proof are reported in Appendix G.1.

## A  Generalization of the Model by Parts

In this section we introduce a slight generalization of the model considered in this work and that will be used in the rest of the appendixes. In particular we consider the case where $P$ is not necessarily finite and, possibly, the observed parts of $y$ are not necessarily deterministic.

### A.1  When the Parts don't correspond exactly

In general, $y_p$ (the $p$-th part of $y$) could not be univocally determined given $p \in P$. For instance, consider a speech recognition problem where the goal is to predict the sentence pronounced by a speaker from an audio signal. In this setting the input space $X$ is the set of all audio signals and $Y = Z$ is the set of all strings that can be produced in the speaker's language. In principle, for any part $x_p$ of an input signal $x \in X$ it is possible to identify the corresponding part $y_p$ of the target string. In practice, such a procedure would require significant preprocessing (e.g. using hidden markov models) and would however not be guaranteed to be error-free.

In general, given an input $x \in X$ a label $y \in Y$ and a part $p \in P$, observations for the $p$-th part of $y$ can be distributed according to some probability $\mu(w|y,x,p)$ over the set $[Y]$ of parts of $Y$. A possible way to model this situation is to consider a characterization of $L$ in terms of a further function $\ell : Z \times [Y] \times X \times P \to \mathbb{R}$ such that

$$\triangle(z,y|x) = \int_P L(z,y|x,p)d\pi(p|x), \quad \text{where} \tag{22}$$

$$L(z,y|x,p) = \int_{[Y]} \ell(z,\eta|x,p) \, d\mu(\eta|y,x,p). \tag{23}$$

In this sense, the distribution $\mu$ can be interpreted as characterizing how likely it is for the part $p$ of an input $x$ with associated label $y$ to correspond to $\eta \in [Y]$. It is possible to recover the standard characterization by selecting $\mu$ to be the Dirac de

$$\mu(\eta|y,x,p) = \delta(\eta,y_p).$$

**Remark 3** (Connection with standard Structured Prediction). *Note that the loss above generalizes the standard structured prediction framework as in [43, 29, 8]. Indeed, it is always possible to formulate a structured prediction loss $\triangle$ in the proposed setting, by taking $\ell = \triangle$ and $P = \{0\}$, $[Y] = Y$, $\pi(0|x) = 1$ and $\mu(w|y,x,0) = \delta_y$. However, if there exists a non-trivial characterization of $\triangle$ in terms of these objects, then the algorithm proposed in this work is able to exploit this additional structure to achieve improved generalization performance.*

Here we give the extended defintion of the SELF assumption, given the definition of loss in (22).

**Definition 2** (SELF by Parts (Extended)). *A function $\triangle : Z \times Y \times X \to \mathbb{R}$ is a* Structure Encoding Loss Function (SELF) by Parts *if it admits a factorization in the form of* (22) *with functions $\ell : Z \times [Y] \times X \times P \to \mathbb{R}$, and there exists a separable Hilbert space $\mathcal{H}$ and two bounded continuous maps $\psi : [Z] \times [X] \times P \to \mathcal{H}$, $\varphi : [Y] \to \mathcal{H}$ such that for any $z \in Z$, $\eta \in [Y]$, $x \in X$, $p \in P$*

$$\ell(z,\eta|x,p) = \langle \psi(z,x,p), \varphi(\eta) \rangle_{\mathcal{H}}. \tag{24}$$

**Remark 4** (Def. 2 is more general than Def. 1). *Given a loss $\triangle$ satisfying Def. 1 for some $\psi', \phi, \mathcal{H}'$, then it satisfy Def. 2, with $\psi(z,x,p) = \psi'(z_p, z_p, p)$, with $\phi = \phi'$ with $\mathcal{H} = \mathcal{H}'$.*

# B Notation and Main Definitions

Let $L^2(X \times P, \pi\rho_X)$ be the Lebesgue function space with norm

$$\|\beta\|^2_{L^2(X \times P, \pi\rho_X)} = \int_{X \times P} \beta(x,p)^2 \, d\pi(p|x)d\rho_X(x)$$

with $\beta : X \times P \to \mathbb{R}$. Analogously, $L^2(X \times P, \pi\rho_X, \mathcal{H})$ be the Lebesgue function space with norm

$$\|\beta\|^2_{L^2(X \times P, \pi\rho_X, \mathcal{H})} = \int_{X \times P} \|\beta(x,p)\|^2_{\mathcal{H}} \, d\pi(p|x)d\rho_X(x)$$

with $\beta : X \times P \to \mathcal{H}$. Let $\big((x_i, y_i)\big)_{i=1}^n$ be the training set and let $\big((x_{i_j}, y_{i_j}, p_j, w_j)\big)_{j=1}^m$. Denote with $\widehat{\rho}_X$ the probability measure $\frac{1}{n}\sum_{i=1}^n \delta_{x_i}$. We define $L^2(X \times P, \pi\widehat{\rho}_X, \mathcal{H})$ the Lebesgue function space with norm

$$\|\beta\|^2_{L^2(X \times P, \pi\widehat{\rho}_X, \mathcal{H})} = \frac{1}{n}\sum_{i=1}^n \int_P \|\beta(x_i, p)\|^2_{\mathcal{H}} \, d\pi(p|x_i).$$

with $\beta : X \times P \to \mathcal{H}$.

Let $k : (X \times P) \times (X \times P) \to \mathbb{R}$ be a reproducing kernel with associated reproducing kernel Hilbert space (RKHS) $\mathcal{F}$. For any $(x, p) \in X \times P$ we denote $k_{x,p} = k\big((x,p), \cdot\big) \in \mathcal{F}$.

We introduce the following objects:

- $S : \mathcal{F} \to L^2(X \times P, \pi\rho_X)$ the operator such that, for any $f \in \mathcal{F}$,

$$(Sf)(\cdot, \cdot) = \big\langle f, k_{(\cdot, \cdot)} \big\rangle_{\mathcal{F}}.$$

- $S^* : L^2(X \times P, \pi\rho_X) \to \mathcal{F}$ the operator such that, for any $\beta \in L^2(X \times P, \pi\rho_X)$,

$$S^*\beta = \int_{X \times P} k_{x,p}\beta(x,p) \, d\pi(p|x)d\rho_X(x).$$

- $C : \mathcal{F} \to \mathcal{F}$ the operator $C = \int_{X \times P} k_{x,p} \otimes k_{x,p} \, d\pi(p|x)d\rho_X(x).$

- $\widetilde{C} : \mathcal{F} \to \mathcal{F}$ the operator $\widetilde{C} = \frac{1}{n}\sum_{i=1}^n \int_P k_{x_i,p} \otimes k_{x_i,p} \, d\pi(p|x_i).$

- $\widehat{C} : \mathcal{F} \to \mathcal{F}$ the operator $\widehat{C} = \frac{1}{m}\sum_{j=1}^m k_{x_{i_j},p_j} \otimes k_{x_{i_j},p_j}.$

- $L : L^2(X \times P, \pi\rho_X) \to L^2(X \times P, \pi\rho_X)$ the operator such that for any $\beta \in L^2(X \times P, \pi\rho_X)$, we have that $(L\beta)(\cdot) = \int_{X \times P} k\big((x,p), \cdot\big)\beta(x,p) \, d\pi(p|x)d\rho_X(x).$

- $B : \mathcal{H} \to \mathcal{F}$ the operator $B = \int_{P \times X} k_{x,p} \otimes \varphi(w) \, d\mu(w|y,x,p)d\pi(p|x)d\rho(y,x)$. Note that by definition $B = \int k_{x,p} \otimes g^*(x,p) \, d\pi(p|x)d\rho_X(x)$ with $g^*$ defined as in (13).

- $\widehat{B} : \mathcal{H} \to \mathcal{F}$ the operator $\widehat{B} = \frac{1}{m}\sum_{j=1}^m k_{x_{i_j},p_j} \otimes \varphi(w_j).$

- $G : \mathcal{H} \to L^2(X \times P, \pi\rho_X)$ the operator such that, for any $h \in \mathcal{H}$ is such that $(Gh)(\cdot) = \big\langle g^*(\cdot), h \big\rangle_{\mathcal{H}}$ for any $h \in \mathcal{H}$, with $g^*$ defined as in (13).

**Further Notation.** Let $\mathcal{H}$ and $\mathcal{F}$ be two Hilbert spaces and let $h \in \mathcal{H}$ and $f \in \mathcal{F}$, we denote with $h \otimes f$ the bounded linear operator from $\mathcal{F} \to \mathcal{H}$ such that, for any $g \in \mathcal{F}$, we have $(h \otimes f)g = h \langle f, g \rangle_{\mathcal{F}}$. Note that $h \otimes f \in \mathcal{H} \otimes \mathcal{F}$, where $\mathcal{H} \otimes \mathcal{F}$ is the tensor product between the Hilbert spaces $\mathcal{H}, \mathcal{F}$ and is isometric to the the space of Hilbert-Schmidt operators from $\mathcal{F}$ to $\mathcal{H}$, denoted by $\mathrm{HS}(\mathcal{F}, \mathcal{H})$, namely the bounded linear operators $G : \mathcal{F} \to \mathcal{H}$ with finite Hilbert-Schmidt norm $\|G\|_{\mathrm{HS}} = \sqrt{\mathrm{Tr}(G^*G)}$.

### B.1 Auxiliary Results

**Lemma 6.** *With the notation introduced above, the following equations hold.*

- $L = SS^*$.

- $C = S^*S$.

- $SC_\lambda^{-1}S^* = LL_\lambda^{-1} = I - \lambda L_\lambda^{-1}$.

- $C_\lambda^{-1}S^* = S^*L_\lambda^{-1}$.

- $\|C_\lambda^{-1/2}S^*\| = \|S^*L_\lambda^{-1/2}\| \leqslant 1$ *for any* $\lambda > 0$

The proof of the result above are well known and we refer to Appendix B in [8] for a proof with same notation as the one adopted in this paper. Below we show two further results that we will need

**Lemma 7.** *with the notation introduced above we have*

$$B = S^*G. \tag{25}$$

*Proof.* By applying the definition of the two operators $S$ and $G$ we have that for any $h \in \mathcal{H}$,

$$S^*Gh = S^*\big((Gh)(\cdot)\big) \tag{26}$$
$$= S^*\big(\langle g^*(\cdot), h \rangle_{\mathcal{H}}\big) \tag{27}$$
$$= \int k_{x,p}\, \langle g^*(x,p), h \rangle_{\mathcal{H}}\ d\pi(p|x)d\rho_X(x) \tag{28}$$
$$= \int (k_{x,p} \otimes g^*(x,p))h\ d\pi(p|x)d\rho_X(x) = Bh \tag{29}$$

Hence $B = S^*G$ as required. $\qquad\square$

## C   Derivation of the algorithm

In this section we show how the algorithm naturally derives from the definition of the problem and in particular we prove Lemma 2. Our analysis starts from the observation that when the loss function is SELF the solution of the learning problem in (5) is completely characterized in terms of the *conditional expectation* of $\varphi(y_p)$ given $x$, denoted by $g^* : X \times P \to \mathcal{H}$, with

$$g^*(x,p) = \int \varphi(\eta)d\mu(\eta|x,y,p)d\rho(y|x). \tag{30}$$

Note that since $\varphi(\cdot)$ is bounded and continuous, we have that $g^* \in L^2(X, \pi\rho_X, \mathcal{H})$. Below we prove Lemma 2

**Lemma 2.** *Let* $\triangle$ *be SELF and* $Z$ *compact. Then, the minimizer of* (5) *is* $\rho_X$*-a.e. characterized by*

$$f^*(x) = \operatorname*{argmin}_{z \in Z} \sum_{p \in P} \pi(p|x) \langle \psi(z_p, x_p, p), g^*(x,p) \rangle_{\mathcal{H}}, \qquad g^*(x,p) = \int_Y \varphi(y_p)d\rho(y|x). \tag{13}$$

*Proof.* By Berge maximum theorem[2] (see also [8]), since $Z$ is compact, we have that the solution of the learning problem in (5) is characterized by

$$f^*(x) = \operatorname*{argmin}_{z \in Z} \int \triangle(z, y|x)d\rho(y|x).$$

The result is obtained by expanding the definition of $\triangle$ with respect to SELF (Def. 2) and the linearity of the inner product and the integral

$$\int \triangle(z,y|x)d\rho(y|x) = \int \ell(z,\eta|x,p)d\mu(\eta|y,x,p)d\pi(p|x)d\rho(y|x) \tag{31}$$

$$= \int \langle \psi(z,x,p), \varphi(\eta)\rangle_{\mathcal{H}} \, d\mu(\eta|y,x,p)d\pi(p|x)d\rho(y|x) \tag{32}$$

$$= \int \left\langle \psi(z,x,p), \int \varphi(\eta)d\mu(\eta|y,x,p)d\rho(y|x) \right\rangle_{\mathcal{H}} d\pi(p|x) \tag{33}$$

$$= \int \langle \psi(z,x,p), g^*(x,p)\rangle_{\mathcal{H}} \, d\pi(p|x), \tag{34}$$

as desired. $\qquad\square$

Since $g^*$ depends on the unknown distribution $\rho$, we substitute it in (13) with an approximation $\widehat{g}$. In particular, since $g^*$ is the conditional expectation induced by $\rho(y|x)$, a viable choice for $\widehat{g}$ is the *empirical risk minimizer* of the squared loss, which is a well known estimator for the conditional expectation [7], namely

$$\widehat{g} = \operatorname*{argmin}_{g \in \mathcal{G}} \frac{1}{m} \sum_{j=1}^{m} \|\psi(\eta_j) - g(\chi_j, p_j)\|_{\mathcal{H}}^2 + \lambda\|g\|_{\mathcal{G}}^2, \tag{35}$$

where $\mathcal{G}$ is a normed space of functions from $X \times P$ to $\mathcal{H}$. In this work we will consider $\mathcal{G} = \mathcal{H} \otimes \mathcal{F}$ where $\mathcal{F}$ is the space of functions associated to a kernel $k$ on $X \times P$. In this case $\widehat{g}$ can be obtained in closed form in terms of the auxiliary dataset and, when plugged in (13), the resulting estimator corresponds exactly to the one in (7), as shown in next Lemma.

**Lemma 8.** *Let $\triangle$ be SELF, $Z$ a compact set and $k$ be a positive definite kernel on $X \times P$ and $\widehat{f}$ defined as in (7) with weights as in (10) computed using kernel $k$. Then $\widehat{f}$ is characterized by*

$$\widehat{f}(x) = \operatorname*{argmin}_{z \in Z} \sum_{p \in P} \pi(p|x) \langle \psi(z_p, x_p, p), \widehat{g}(x,p)\rangle_{\mathcal{H}}, \tag{36}$$

*with $\widehat{g}$ the solution of (35) computed using kernel $k$.*

*Proof.* We recall (see [7]) that the least-squares solution of (35) can be obtained in close form solution as

$$\widehat{g}(x,p) = \sum_{j=1}^{m} \alpha_j(x,p)\varphi(y_{p_j})$$

for any $x \in X$ and $p \in P$, where the weights $\alpha$ are defined as in (10). By linearity of the inner product we have

$$\sum_{p \in P} \pi(p|x) \langle \psi(z_p, x_p, p), \widehat{g}(x,p)\rangle_{\mathcal{H}} = \sum_{j=1}^{m}\sum_{p \in P} \pi(p|x)\alpha_j(x,p) \langle \psi(z_p, x_p, p), \varphi(y_{p_j})\rangle_{\mathcal{H}} \tag{37}$$

$$= \sum_{j=1}^{m}\sum_{p \in P} \pi(p|x)\alpha_j(x,p)L_p(z_p, y_p|x_p) \tag{38}$$

where the last step follows from the assumption that the loss is SELF. $\qquad\square$

An interesting consequence of the lemma above is that $\psi, \varphi, \widehat{g}, g^*, \mathcal{H}$ are only needed for theoretical purposes – i.e. to establish the connection between the estimator $\widehat{f}$ and the ideal solution $f^*$ – and are not needed for the evaluation of $\widehat{f}$ which is done in terms of known objects, via (7).

# D Comparison Inequality

In this we derive a result, Thm. 9, that is crucial to prove the statistical properties of the proposed algorithm. Note that it is analogous to the Comparison Inequality of [8] and of independent interest for the proposed framework. First we define the following estimator, that is a more general version of the one presented in the paper

$$\widehat{f}(x) = \operatorname*{argmin}_{z \in Z} \int_P \langle \psi(z, x, p), \widehat{g}(x, p) \rangle_{\mathcal{H}} \, \pi(p|x). \tag{39}$$

Note that the estimator presented in the main paper which is characterized by (36), Lemma 8 can be written like (39), applying Remark 4 in Appendix A.1.

**Theorem 9.** *When $Z$ is a compact set and $\triangle$ satisfies Def. 2, for any measurable $\widehat{g} : X \times P \to \mathcal{H}$ and $\widehat{f} : X \to Z$ defined in terms of $\widehat{g}$ as in (39). Then*

$$\mathcal{E}(\widehat{f}) - \mathcal{E}(f^*) \leqslant c_\triangle \|\widehat{g} - g^*\|_{L^2(X \times P, \pi \rho_X, \mathcal{H})} \tag{40}$$

*and $c_\triangle$ is a constant depending only on $\triangle$ and defined at the end of the proof.*

*Proof.* For any $x \in X$ and $z \in Z$, let

$$A(z|x) = \int_P \langle \psi(z, x, p), g^*(x, p) \rangle_{\mathcal{H}} \, d\pi(p|x), \tag{41}$$

$$\widehat{A}(z|x) = \int_P \langle \psi(z, x, p), \widehat{g}(x, p) \rangle_{\mathcal{H}} \, d\pi(p|x). \tag{42}$$

By the SELF assumption $\ell(z, w|x, p) = \langle \psi(z, x, p), \varphi(w) \rangle_{\mathcal{H}}$ and the definition of $g^*$ as in (13) we have the following alternative characterization for $A(z|x)$ as shown in Lemma 2

$$A(z|x) = \int_{[Y] \times Y \times P} \ell(z, w|x, p) \, d\mu(w|y, x, p) d\rho(y|x) d\pi(p|x). \tag{43}$$

Then, $\mathcal{E}(f) = \int_X A(f(x)|x) \, d\rho_X(x)$ for any $f : X \to Z$ and we have the following decomposition of the excess risk

$$\mathcal{E}(\widehat{f}) - \mathcal{E}(f^*) = \int_X A(\widehat{f}(x)|x) - A(f^*(x)|x) \, d\rho_X(x) \tag{44}$$

$$= \int_X A(\widehat{f}(x)|x) - \widehat{A}(\widehat{f}(x)|x) + \underbrace{\widehat{A}(\widehat{f}(x)|x) - \widehat{A}(f^*(x)|x)}_{\leqslant 0} \tag{45}$$

$$+ \int_X \widehat{A}(f^*(x)|x) - A(f^*(x)|x) \, d\rho_X(x) \tag{46}$$

$$\leqslant 2 \int_X \sup_{z \in Z} \left| \widehat{A}(z|x) - A(z|x) \right| \, d\rho_X(x) \tag{47}$$

where we have used the fact that $\widehat{A}(\widehat{f}(x)|x) - \widehat{A}(f^*(x)|x) \leqslant 0$ since, by definition, $\widehat{f}(x)$ is the minimizer of $\widehat{A}(\cdot|x)$ (see Eq. (39)).

Now, note that by the linearity of the inner product we have

$$\left| \widehat{A}(z|x) - A(z|x) \right| = \left| \int_P \langle \psi(z, x, p), \widehat{g}(x, p) - g^*(x, p) \rangle_{\mathcal{H}} \, d\pi(p|x) \right| \tag{48}$$

$$\leqslant \int_P \|\psi(z, x, p)\|_{\mathcal{H}} \, \|g^*(x, p) - \widehat{g}(x, p)\|_{\mathcal{H}} \, d\pi(p|x) \tag{49}$$

$$\leqslant \sqrt{\int_P \|\psi(z, x, p)\|_{\mathcal{H}}^2 \, d\pi(p|x)} \sqrt{\int_P \|g^*(x, p) - \widehat{g}(x, p)\|_{\mathcal{H}}^2 \, d\pi(p|x)} \tag{50}$$

$$= q(x, z) \sqrt{\int_P \|g^*(x, p) - \widehat{g}(x, p)\|_{\mathcal{H}}^2 \, d\pi(p|x)} \tag{51}$$

where we applied Cauchy-Schwartz for each of the two inequalities, with $q(x, z) = \sqrt{\int_P \|\psi(z, x, p)\|_{\mathcal{H}}^2 \, d\pi(p|x)}$.

Denote with $\| \cdot \|_{L^2(X \times P, \pi\rho_X, \mathcal{H})}$ the norm such that

$$\|g\|_{L^2(X \times P, \pi\rho_X, \mathcal{H})}^2 = \int_{X \times P} \|g(x, p)\|_{\mathcal{H}}^2 \, d\pi(p|x) d\rho_X(x), \tag{52}$$

for any $g : X \times P \to \mathcal{H}$. Then, plugging the inequality above in (47), we obtain

$$2 \int_X \sup_{z \in Z} \left| \widehat{A}(z|x) - A(z|x) \right| d\rho_X(x) \tag{53}$$

$$\leqslant 2 \int_X \sup_{z \in Z} \left[ q(x, z) \sqrt{\int_P \|g^*(x, p) - \widehat{g}(x, p)\|_{\mathcal{H}}^2 \, d\pi(p|x)} \right] d\rho_X(x) \tag{54}$$

$$= 2 \int_X \sup_{z \in Z} \left[ q(x, z) \right] \sqrt{\int_P \|g^*(x, p) - \widehat{g}(x, p)\|_{\mathcal{H}}^2 \, d\pi(p|x)} \, d\rho_X(x) \tag{55}$$

$$\leqslant 2 \sqrt{\int_X \left( \sup_{z \in Z} q(x, z) \right)^2 d\rho_X(x)} \sqrt{\int_{X \times P} \|g^*(x, p) - \widehat{g}(x, p)\|_{\mathcal{H}}^2 \, d\pi(p|x) d\rho_X(x)} \tag{56}$$

$$= c_{\triangle} \|\widehat{g} - g^*\|_{L^2(X \times P, \pi\rho_X, \mathcal{H})} \tag{57}$$

where the last inequality follows from Cauchy-Schwartz and

$$c_{\triangle} = 2 \sqrt{\int_X \left( \sup_{z \in Z} q(x, z) \right)^2 d\rho_X(x)} \tag{58}$$

$$= 2 \sqrt{\int_X \sup_{z \in Z} \left[ \int_P \|\psi(z, x, p)\|_{\mathcal{H}}^2 \, d\pi(p|x) \right] d\rho_X(x)} \tag{59}$$

$\square$

**Remark 5** (Remove the dependency of $c_{\triangle}$ from $\rho_X$)**.** *Note that it is always possible to remove the dependency of $c_{\triangle}$ from $\rho_X$ by bounding it with*

$$c_{\triangle} \leqslant 2 \left( \sup_{\substack{z \in Z \\ x \in X}} \int_P \|\psi(z, x, p)\|_{\mathcal{H}}^2 \, d\pi(p|x) \right)^{1/2} \tag{60}$$

# E  Analytical Decomposition

According to the comparison inequality (40) it is sufficient to bound the quantity $\|\widehat{g} - g^*\|_{L^2(X \times P, \pi\rho_X, \mathcal{H})}$ in order to control the excess risk of the estimator $\widehat{f}$. Equipped with the notation introduced above, we can now focus on studying this quantity. In particular in Thm. 11 we provide an analytical decomposition of $\|\widehat{g} - g^*\|_{L^2(X \times P, \pi\rho_X, \mathcal{H})}$ in terms of basic quantities that can be controlled in expectation (or probability, for the universal consistency).

**Proposition 10.** *Let $\widehat{g}, g^*$ be defined as in (35) and (30), then the following holds*

$$\|\widehat{g} - g^*\|_{L^2(X \times P, \pi\rho_X, \mathcal{H})} = \|S\widehat{C}_\lambda^{-1}\widehat{B} - G\|_{\mathrm{HS}(\mathcal{H}, L^2(X \times P, \pi\rho_X))} \tag{61}$$

*Proof.* First of all we recall that the space $L^2(X \times P, \pi\rho_X, \mathcal{H})$ is isometric to $\mathcal{H} \otimes L^2(X \times P, \pi\rho_X)$ which is isometric to the space of linear Hilbert-Schmidt operators from $\mathcal{H} \to L^2(X \times P, \pi\rho_X)$, denoted by $\mathrm{HS}(\mathcal{H}, L^2(X \times P, \pi\rho_X))$. Now note that $G$ is the operator in $\mathrm{HS}(\mathcal{H}, L^2(X \times P, \pi\rho_X))$, that is isometric to $g^* \in L^2(X \times P, \pi\rho_X, \mathcal{H})$, indeed $Gv = \langle g^*(\cdot, \cdot), v \rangle_{\mathcal{H}}$, for any $v \in \mathcal{H}$.

Now note that is the solution of the problem in (35). Indeed, first note that the functional $\widehat{R}_\lambda(W)$, defining the problem in (35), is smooth and strongly convex ($W \in \mathcal{H} \otimes \mathcal{F}, \lambda > 0$). Then we find the

solution by equating the derivative of $\widehat{R}_\lambda(W)$ to 0. First note that for any $W \in \mathcal{H} \otimes \mathcal{F}$, the functional $\widehat{R}_\lambda(W)$, is equivalent to

$$\widehat{R}_\lambda(W) = \frac{1}{m} \sum_{j=1}^{m} \|\phi(w_j) - W k_{(x_{i_j}, p_j)}\|_{\mathcal{H}}^2 + \lambda \|W\|_{\mathcal{H} \otimes \mathcal{F}} \tag{62}$$

$$= \mathrm{Tr}\Big[ W \left( \frac{1}{m} \sum_{j=1}^{m} k_{(x_{i_j}, p_j)} \otimes k_{(x_{i_j}, p_j)} + \lambda I \right) W^* \tag{63}$$

$$- 2 \left( \frac{1}{m} \sum_{j=1}^{m} k_{(x_{i_j}, p_j)} \otimes \phi(w_j) \right) W^* + \frac{1}{m} \sum_{j=1}^{m} \phi(w_j) \otimes \phi(w_j) \Big] \tag{64}$$

$$= \mathrm{Tr}\Big[ W \left( \widehat{C} + \lambda I \right) W^* - 2 \widehat{B} W + \frac{1}{m} \sum_{j=1}^{m} \phi(w_j) \otimes \phi(w_j) \Big], \tag{65}$$

where for the last step we applied the defintion of $\widehat{C}$ and $\widehat{B}$. By taking the derivative of $\widehat{R}_\lambda(W)$ in $W$ and equating it to 0 the following minimizer is obtained $\widehat{W} = \widehat{B}^* \widehat{C}_\lambda^{-1}$.

Moreover note that, $S\widehat{C}_\lambda^{-1}\widehat{B}$ is the operator in $\mathrm{HS}(\mathcal{H}, L^2(X \times P, \pi\rho_X))$, that is isometric to $\widehat{g} \in L^2(X \times P, \pi\rho_X, \mathcal{H})$, indeed by definition of $S$

$$S\widehat{C}_\lambda^{-1}\widehat{B} v = \left\langle k_{(\cdot, \cdot)}, \widehat{W}^* v \right\rangle_{\mathcal{F}} = \left\langle \widehat{W} k_{(\cdot, \cdot)}, v \right\rangle_{\mathcal{H}} = \langle \widehat{g}(\cdot, \cdot), v \rangle_{\mathcal{H}}, \quad \forall v \in \mathcal{H}.$$

$\square$

**Theorem 11.** *Let $\lambda > 0$. With the definitions in [Appendix B](#), we have*

$$\|\widehat{g} - g^*\|_{L^2(X \times P, \pi\rho_X, \mathcal{H})} \leqslant \left( \frac{1}{\sqrt{\lambda}} + \frac{\beta_1^{1/2}}{\lambda} \right) \left( \beta_1 \mathcal{A}_{1/2}(\lambda) + \beta_2 \right) + \lambda \mathcal{A}_1(\lambda). \tag{66}$$

*where $\beta_1 = \|C - \widehat{C}\|$, $\beta_2 = \|\widehat{B} - B\|_{\mathrm{HS}}$ and $\mathcal{A}_r(\lambda) = \|L_\lambda^{-r} G\|_{\mathrm{HS}}$ for $r > 0$.*

*Proof.* By [Prop. 10](#) and by adding and subtracting $S\widehat{C}_\lambda^{-1}B$ and $SC_\lambda^{-1}B$ we have

$$\|\widehat{g} - g^*\|_{L^2(X \times P, \pi\rho_X, \mathcal{H})} = \|S\widehat{C}_\lambda^{-1}\widehat{B} - G\|_{\mathrm{HS}(\mathcal{H}, L^2)} \leqslant A_1 + A_2 + A_3 \tag{67}$$

with

$$A_1 = \|S\widehat{C}_\lambda^{-1}\widehat{B} - S\widehat{C}_\lambda^{-1}B\|_{\mathrm{HS}(\mathcal{H}, L^2)} \tag{68}$$

$$A_2 = \|S\widehat{C}_\lambda^{-1}B - SC_\lambda^{-1}B\|_{\mathrm{HS}(\mathcal{H}, L^2)} \tag{69}$$

$$A_3 = \|SC_\lambda^{-1}B - G\|_{\mathrm{HS}(\mathcal{H}, L^2)}. \tag{70}$$

**Bounding $A_1$.** Now, by dividing and multiplying by $C_\lambda^{1/2}$, we have

$$A_1 = \|S\widehat{C}_\lambda^{-1}(\widehat{B} - B)\|_{\mathrm{HS}(\mathcal{H}, L^2)} \leqslant \|S\widehat{C}_\lambda^{-1}\| \|\widehat{B} - B\|_{\mathrm{HS}(\mathcal{H}, \mathcal{F})} \tag{71}$$

**Bounding $A_2$.** By using the identity $R^{-1} - T^{-1} = R^{-1}(T - R)T^{-1}$ holding for any invertible operators $R, T : \mathcal{F} \to \mathcal{F}$, we have

$$A_2 = \|S(\widehat{C}_\lambda^{-1} - C_\lambda^{-1})B\|_{\mathrm{HS}(\mathcal{H}, L^2)} \tag{72}$$

$$= \|S\widehat{C}_\lambda^{-1}(C_\lambda - \widehat{C}_\lambda)C_\lambda^{-1}B\|_{\mathrm{HS}(\mathcal{H}, L^2)} \tag{73}$$

$$= \|S\widehat{C}_\lambda^{-1}(C - \widehat{C})C_\lambda^{-1}B\|_{\mathrm{HS}(\mathcal{H}, L^2)} \tag{74}$$

$$\leqslant \|S\widehat{C}_\lambda^{-1}\| \|C - \widehat{C}\| \|C_\lambda^{-1}B\|_{\mathrm{HS}(\mathcal{H}, \mathcal{F})}. \tag{75}$$

$$\tag{76}$$

We further apply Lemma 6 to have $\|C_\lambda^{-1/2}S^*\| = \|S^*L_\lambda^{-1/2}\| \leqslant 1$ and $C_\lambda^{-1}S = S^*L_\lambda^{-1}$. Then,

$$\|C_\lambda^{-1}B\|_{\mathrm{HS}(\mathcal{H},\mathcal{F})} = \|C_\lambda^{-1}S^*G\|_{\mathrm{HS}(\mathcal{H},\mathcal{F})} = \|S^*L_\lambda^{-1}G\|_{\mathrm{HS}(\mathcal{H},\mathcal{F})} \tag{77}$$

$$\leqslant \|S^*L_\lambda^{-1/2}\|\|L_\lambda^{-1/2}G\|_{\mathrm{HS}(\mathcal{H},L^2)} \leqslant \|L_\lambda^{-1/2}G\|_{\mathrm{HS}(\mathcal{H},L^2)}. \tag{78}$$

**Bounding $A_3$.** From Lemma 6 we have $B = S^*G$ and $SC_\lambda^{-1}S^* = LL_\lambda^{-1} = I - \lambda L_\lambda^{-1}$. Then,

$$A_3 = \|SC_\lambda^{-1}S^*G - G\|_{\mathrm{HS}(\mathcal{H},L^2)} = \|(I - \lambda L_\lambda^{-1})G - G\|_{\mathrm{HS}(\mathcal{H},L^2)} = \lambda\|L_\lambda^{-1}G\|_{\mathrm{HS}(\mathcal{H},L^2)}. \tag{79}$$

To conclude, we control the term $\|S\widehat{C}_\lambda^{-1}\|$ by

$$\|S\widehat{C}_\lambda^{-1}\|^2 = \|\widehat{C}_\lambda^{-1}C\widehat{C}_\lambda^{-1}\| \leqslant \|\widehat{C}_\lambda^{-1}(C - \widehat{C})\widehat{C}_\lambda^{-1}\| + \|\widehat{C}_\lambda^{-1}\widehat{C}\widehat{C}_\lambda^{-1}\| \tag{80}$$

$$\leqslant \|\widehat{C}_\lambda^{-1}\|^2\|C - \widehat{C}\| + \frac{1}{\lambda} \tag{81}$$

$$\leqslant \frac{1}{\lambda^2}\|C - \widehat{C}\| + \frac{1}{\lambda} \tag{82}$$

Therefore

$$\|S\widehat{C}_\lambda^{-1}\| \leqslant \sqrt{\frac{\|C - \widehat{C}\|}{\lambda^2} + \frac{1}{\lambda}} \leqslant \frac{1}{\sqrt{\lambda}} + \frac{\sqrt{\|C - \widehat{C}\|}}{\lambda} \tag{83}$$

Combining the bounds for $A_1$, $A_2$ and $A_3$ we obtain the desired result. $\qquad\square$

# F    Learning Rates

Building on the analytic decomposition of Thm. 11 we observe that the key quantities to study in this setting are the $\mathbb{E}\|\widehat{C} - C\|^2$ and $\mathbb{E}\|\widehat{B} - B\|_{\mathrm{HS}}^2$ as discussed below. In particular the following theorem further decomposes the quantities from Thm. 11, and $\mathbb{E}\|\widehat{C} - C\|^2$ and $\mathbb{E}\|\widehat{B} - B\|_{\mathrm{HS}}^2$, are bounded in Appendices F.1 and F.2. Finally Thm. 20 is given in Appendix F.3.

**Theorem 12.** *Let $\lambda > 0$. With the definitions in Appendix B and Thm. 11, we have*

$$\mathbb{E}\|\widehat{g} - g^*\|_{L^2(X \times P, \pi\rho_X, \mathcal{H})} \leqslant 2\left(1 + \frac{\sqrt{\mathbb{E}\beta_1^2}}{\lambda}\right)^{1/2}\left(\frac{\mathcal{A}_{1/2}(\lambda)^2\mathbb{E}\beta_1^2}{\lambda} + \frac{\mathbb{E}\beta_2^2}{\lambda}\right)^{1/2} + \lambda\mathcal{A}_1(\lambda). \tag{84}$$

*Proof.* Let $a = \frac{1}{\sqrt{\lambda}}$, $b = \frac{1}{\lambda}$, $c = \|L_\lambda^{-1/2}G\|_{\mathrm{HS}}$ and $d = \lambda\|L_\lambda^{-1}G\|_{\mathrm{HS}}$. Then,

$$\mathbb{E}\|\widehat{g} - g^*\|_{L^2(X \times P, \pi\rho_X, \mathcal{H})} \leqslant \mathbb{E}(a + b\beta_1^{1/2})(c\beta_1 + \beta_2) + d \tag{85}$$

$$\leqslant \sqrt{\mathbb{E}(a + b\beta_1^{1/2})^2\mathbb{E}(c\beta_1 + \beta_2)^2} + d \tag{86}$$

$$\leqslant \sqrt{4(a^2 + b^2\mathbb{E}\beta_1)(c^2\mathbb{E}\beta_1^2 + \mathbb{E}\beta_2^2)} + d \tag{87}$$

$$\leqslant 2\sqrt{(a^2 + b^2\sqrt{\mathbb{E}\beta_1^2})(c^2\mathbb{E}\beta_1^2 + \mathbb{E}\beta_2^2)} + d, \tag{88}$$

as desired. $\qquad\square$

The rest of this section will be devoted to characterizing the behavior of $\mathbb{E}\beta_1^2$ and $\mathbb{E}\beta_2^2$ in order to obtain a more interpretable learning rates for the estimator proposed in this work.

## F.1    Bounding $\mathbb{E}\beta_1^2$

Denote $\zeta_{x_{i_j},p_j} = k_{x_{i_j},p_j} \otimes k_{x_{i_j},p_j} - C$. First, we show that $\mathbb{E}\zeta_{x_{i_j},p_j} = 0$.

**Lemma 13.** *With the definition above, when $x_1, \ldots, x_n$ are identically distributed, we have*

$$\mathbb{E}\,\zeta_{x_{i_j},p_j} = 0$$

*Proof.* Since $x_1, \ldots, x_n$ are identically distributed, for any $j = 1, \ldots, m$, we have

$$\mathbb{E}\, k_{x_{i_j}, p_j} \otimes k_{x_{i_j}, p_j} = \frac{1}{n} \sum_{i_j=1}^{n} \int_{P \times X} k_{x_{i_j}, p_j} \otimes k_{x_{i_j}, p_j}\, d\pi(p_j | x_{i_j}) d\rho_X(x_{i_j}) \tag{89}$$

$$= \int_{P \times X} k_{x,p} \otimes k_{x,p}\, d\pi(p|x) d\rho_X(x) \tag{90}$$

$$= C, \tag{91}$$

as desired $\qquad\square$

**Lemma 14.** *With the definitions of Section B let* $Q_1 = \mathbb{E}\|\zeta_{x,p}\|_{\mathrm{HS}}^2$ *and*

$$\mathfrak{C} = \int_{P \times X} \zeta_{x,p} \zeta_{x,p'}\, d\pi(p|x) d\pi(p'|x) d\rho_X(x) \tag{92}$$

$$\mathbb{E}\|\widehat{C} - C\|_{\mathrm{HS}}^2 = \frac{Q_1}{m} + \frac{(m-1)}{m} \frac{\mathrm{Tr}(\mathfrak{C})}{n}. \tag{93}$$

*Proof.* From the definition of $\widehat{C}$, we have

$$\mathbb{E}\|\widehat{C} - C\|_{\mathrm{HS}}^2 = \mathbb{E}\|\frac{1}{m} \sum_{j=1}^{m} \zeta_{x_{i_j}, p_j}\|_{\mathrm{HS}}^2 = \frac{1}{m^2} \sum_{j,h=1}^{m} \mathbb{E}\, \mathrm{Tr}\left( \zeta_{x_{i_j}, p_j} \zeta_{x_{i_h}, p_h} \right) \tag{94}$$

We consider separately the elements in the sum that correspond to the case $j = h$ and $j \neq h$.

**Case $j = h$.** We have

$$\mathbb{E}\, \mathrm{Tr}\left( \zeta_{x_{i_j}, p_j} \zeta_{x_{i_h}, p_h} \right) = \mathbb{E}\|\zeta_{x_{i_j}, p_j}\|_{\mathrm{HS}}^2 = Q_1 \tag{95}$$

**2. Case $j \neq h$.** We have $\mathbb{E}\, \mathrm{Tr}\left( \zeta_{x_{i_j}, p_j} \zeta_{x_{i_h}, p_h} \right) = \frac{1}{n^2} \sum_{i_j, i_h=1}^{n} R_{i_j, i_h}^{j,h}$ where

$$R_{u,v}^{j,h} = \int_{P \times X} \mathrm{Tr}(\zeta_{x_u, p_j} \zeta_{x_v, p_h})\, d\pi(p_j | x_u) d\pi(p_h | x_v) d\rho_X(x_1) \cdots d\rho_X(x_n). \tag{96}$$

We consider separately the case $i_j = i_h$ and $i_j \neq i_h$.

**Case $j \neq h$ and $i_j = i_h$.** We have that

$$R_{i_j, i_j}^{j,h} = \int_{P \times X} \mathrm{Tr}\left( \zeta_{x_{i_j}, p_j} \zeta_{x_{i_j}, p_h} \right)\, d\pi(p_j | x_{i_j}) d\pi(p_h | x_{i_j}) d\rho_X(x_{i_j}) \tag{97}$$

$$= \int_{P \times X} \mathrm{Tr}\left( \zeta_{x,p} \zeta_{x,p'} \right)\, d\pi(p|x) d\pi(p'|x) d\rho_X(x) = \mathrm{Tr}(\mathfrak{C}). \tag{98}$$

**2.2 Case $j \neq h$ and $i_j \neq i_h$.** We have that

$$R_{i_j, i_h}^{j,h} = \int \mathrm{Tr}\left( \zeta_{x_{i_j}, p_j} \zeta_{x_{i_h}, p_h} \right)\, d\pi(p_j | x_{i_j}) d\pi(p_h | x_{i_h}) d\rho_X(x_{i_j}) d\rho_X(x_{i_h}) \tag{99}$$

$$= \int \mathrm{Tr}\left( \zeta_{x,p} \zeta_{x', p'} \right)\, d\pi(p|x) d\pi(p'|x') d\rho_X(x) d\rho_X(x') \tag{100}$$

$$= \mathrm{Tr}\left( \int \zeta_{x,p}\, d\pi(p|x) d\rho_X(x) \int \zeta_{x', p'}\, d\pi(p'|x') d\rho_X(x') \right) \tag{101}$$

$$= \|\mathbb{E}\, \zeta_{x,p}\|_{\mathrm{HS}}^2 = 0 \tag{102}$$

where the last equality follows from the fact that the $\zeta_{x,p}$ have zero mean according to Lemma 13.

**Combining the above cases.** Note that in (94), Case 1 occurs $m$ times and Case 2 occurs the remaining $m(m-1)$ times. Therefore, we have

$$\mathbb{E}\|\widehat{C} - C\|_{\mathrm{HS}}^2 = \frac{Q_1}{m} + \frac{m-1}{m}\frac{1}{n^2}\sum_{i_j,i_h=1}^{n} R_{i_j,i_h}^{j,h} \tag{103}$$

Now, for the second term on the right hand side, Case 2.1 occurs $n$ times while Case 2.2 occurs the remaining $n(n-1)$ times, leading to the desired result. $\qquad\square$

**Lemma 15.** *With the notation of Lemma 14 and the definition of $\mathsf{q}$ in (167), we have*

$$\mathrm{Tr}(\mathfrak{C}) = \mathfrak{c}_1 - \mathfrak{c}_2 = \mathsf{q}, \tag{104}$$

*where*

$$\mathfrak{c}_1 = \int k\big((x,p),(x,p')\big)^2\, d\pi(p|x)d\pi(p'|x)d\rho_X(x) \tag{105}$$

$$\mathfrak{c}_2 = \int k\big((x,p),(x',p')\big)^2\, d\pi(p|x)d\pi(p'|x')d\rho_X(x)d\rho_X(x'). \tag{106}$$

*Proof.* Note that by definition of $\zeta$ and the reproducing property of the kernel $k$, for any $x,x' \in X$ and $p,p' \in P$ the following holds

$$\mathrm{Tr}(\zeta_{x,p}\zeta_{x',p'}) = k\big((x,p),(x',p')\big)^2 - \mathrm{Tr}\left(C\left(k_{x,p}\otimes k_{x,p}\right)\right) \tag{107}$$

$$- \mathrm{Tr}\left(C\left(k_{x',p'}\otimes k_{x',p'}\right)\right) + \mathrm{Tr}(C^2). \tag{108}$$

Then, by definition of $C = \mathbb{E}\, k_{x,p}\otimes k_{x,p}$, we have

$$\mathrm{Tr}(\mathfrak{C}) = \int \mathrm{Tr}\left(\zeta_{x,p}\zeta_{x,p'}\right)\, d\pi(p|x)d\pi(p'|x)d\rho_X(x) \tag{109}$$

$$= -\mathrm{Tr}(C^2) + \int k\big((x,p),(x,p')\big)^2\, d\pi(p|x)d\pi(p'|x)d\rho_X(x) \tag{110}$$

$$= -\mathrm{Tr}(C^2) + \int k\big((x,p),(x,p')\big)^2\, d\pi(p|x)d\pi(p'|x)d\rho_X(x) \tag{111}$$

$$= \mathfrak{c}_1 - \mathrm{Tr}(C^2). \tag{112}$$

To conclude,

$$\mathrm{Tr}(C^2) = \mathrm{Tr}\left(\left(\int k_{x,p}\otimes k_{x,p}\, d\pi(p|x)d\rho_X(x)\right)\left(\int k_{x',p'}\otimes k_{x',p'}\, d\pi(p'|x')d\rho_X(x')\right)\right) \tag{113}$$

$$= \int k\big((x,p),(x',p')\big)^2\, d\pi(p|x)d\pi(p'|x')d\rho_X(x)d\rho_X(x') \tag{114}$$

$$= \mathfrak{c}_2. \tag{115}$$

The last step consists in noting that $\mathfrak{c}_1 - \mathfrak{c}_2$ is exactly the definition of $\mathsf{q}$ in (167). $\qquad\square$

## F.2 Bounding $\mathbb{E}\beta_2^2$

The analysis for $\mathbb{E}\beta_2^2$ is analogous to that of $\mathbb{E}\beta_1^2$. For completeness we report it below. Denote $\eta_{x_{i_j},p_j,w_j} = k_{x_{i_j},p_j}\otimes\varphi(w_j) - B$. We show that $\mathbb{E}\,\eta_{x_{i_j},p_j,w_j} = 0$.

**Lemma 16.** *With the definition above, when $x_1,\ldots,x_n$ are identically distributed, we have*

$$\mathbb{E}\,\eta_{x_{i_j},p_j,w_j} = 0$$

*Proof.* Since $x_1, \ldots, x_n$ are identically distributed, for any $j = 1, \ldots, m$, we have

$$\mathbb{E}\, k_{x_{i_j}, p_j} \otimes \varphi(w_j) = \frac{1}{n} \sum_{i_j=1}^{n} \int k_{x_{i_j}, p_j} \otimes \varphi(w_j)\, d\mu(w_j | y_{i_j}, x_{i_j}, p_j) d\pi(p_j | x_{i_j}) d\rho(y_{i_j}, x_{i_j}) \quad (116)$$

$$= \int k_{x,p} \otimes \varphi(w)\, d\mu(w | y, x, p) d\pi(p | x) d\rho(y, x) \quad (117)$$

$$= B, \quad (118)$$

as desired. $\qquad\square$

**Lemma 17.** *Let* $Q_2 = \mathbb{E} \|\eta_{x,p,w}\|_{\mathrm{HS}}^2$ *and*

$$\mathfrak{B} = \int \eta_{x,p,w}^* \eta_{x,p',w'}\, d\mu(w | y, x, p) d\mu(w' | y, x, p') d\pi(p | x) d\pi(p' | x) d\rho(y, x) \quad (119)$$

$$\mathbb{E} \|\widehat{B} - B\|_{\mathrm{HS}}^2 = \frac{Q_2}{m} + \frac{(m-1)}{m} \frac{\mathrm{Tr}(\mathfrak{B})}{n}. \quad (120)$$

*Proof.* From the definition of $\widehat{B}$, we have

$$\mathbb{E} \|\widehat{B} - B\|_{\mathrm{HS}}^2 = \mathbb{E} \|\frac{1}{m} \sum_{j=1}^{m} \eta_{x_{i_j}, p_j, w_j}\|_{\mathrm{HS}}^2 = \frac{1}{m^2} \sum_{j,h=1}^{m} \mathbb{E}\, \mathrm{Tr}\left( \eta_{x_{i_j}, p_j, w_j}^* \eta_{x_{i_h}, p_h, w_h} \right) \quad (121)$$

We consider separately the elements in the sum that correspond to the case $j = h$ and $j \neq h$.

**1. Case $j = h$.** We have

$$\mathbb{E}\, \mathrm{Tr}\left( \eta_{x_{i_j}, p_j, w_j}^* \eta_{x_{i_h}, p_h, w_h} \right) = \mathbb{E} \|\eta_{x_{i_j}, p_j, w_j}\|_{\mathrm{HS}}^2 = Q_2. \quad (122)$$

**2. Case $j \neq h$.** We have $\mathbb{E}\, \mathrm{Tr}\left( \eta_{x_{i_j}, p_j, w_j}^* \eta_{x_{i_h}, p_h, w_h} \right) = \frac{1}{n^2} \sum_{i_j, i_h=1}^{n} Z_{i_j, i_h}^{j,h}$ where

$$Z_{u,v}^{j,h} = \int \mathrm{Tr}(\eta_{x_u, p_j, w_j}^* \eta_{x_v, p_h, w_h})\, d\mu(w_j | y_{i_j}, x_{i_j}, p_j) d\mu(w_h | y_{i_h}, x_{i_h}, p_h) \times \quad (123)$$

$$\times\, d\pi(p_j | x_u) d\pi(p_h | x_v) d\rho(y_1, x_1) \cdots d\rho(y_n, x_n). \quad (124)$$

We consider separately the case $i_j = i_h$ and $i_j \neq i_h$.

**2.1 Case $j \neq h$ and $i_j = i_h$.** We have that

$$Z_{i_j, i_j}^{j,h} = \int \mathrm{Tr}\left( \eta_{x_{i_j}, p_j, w_j}^* \eta_{x_{i_j}, p_h, w_h} \right)\, d\mu(w_j | y_{i_j}, x_{i_j}, p_j) d\mu(w_h | y_{i_j}, x_{i_j}, p_h) \times \quad (125)$$

$$\times\, d\pi(p_j | x_{i_j}) d\pi(p_h | x_{i_j}) d\rho(y_{i_j}, x_{i_j}) \quad (126)$$

$$= \int \mathrm{Tr}\left( \eta_{x,p,w}^* \eta_{x,p',w'} \right)\, d\mu(w | y, x, p) d\mu(w' | y, x, p') d\pi(p | x) d\pi(p' | x) d\rho(y, x) \quad (127)$$

$$= \mathrm{Tr}(\mathfrak{B}). \quad (128)$$

**2.2 Case $j \neq h$ and $i_j \neq i_h$.** We have that

$$Z_{i_j,i_h}^{j,h} = \int \mathrm{Tr}\left(\eta_{x_{i_j},p_j,w_j}^* \eta_{x_{i_h},p_h,w_h}\right) \, d\mu(w_j|y_{i_j}, x_{i_j}, p_j) d\mu(w_h|y_{i_h}, x_{i_h}, p_h) \times \tag{129}$$

$$\times \, d\pi(p_j|x_{i_j}) d\pi(p_h|x_{i_h}) d\rho(y_{i_j}, x_{i_j}) d\rho(y_{i_h}, x_{i_h}) \tag{130}$$

$$= \int \mathrm{Tr}\left(\eta_{x,p,w}^* \eta_{x',p',w'}\right) \, d\mu(w|y, x, p) d\mu(w'|y', x', p') \times \tag{131}$$

$$\times \, d\pi(p|x) d\pi(p'|x') d\rho(y, x) d\rho(y', x') \tag{132}$$

$$= \mathrm{Tr}\left(\int \eta_{x,p,w}^* \, d\mu(w|y, x, p) d\pi(p|x) d\rho(y, x) \times \right. \tag{133}$$

$$\left. \times \int \eta_{x',p',w'} \, d\mu(w'|y', x', p') d\pi(p'|x') d\rho(y', x')\right) \tag{134}$$

$$= \|\mathbb{E}\, \eta_{x,p,w}\|_{\mathrm{HS}}^2 = 0, \tag{135}$$

where the last equality follows from the fact that the $\eta_{x,p,w}$ have zero mean according to Lemma 16.

**Combining the above cases.** Note that in (121), Case 1 occurs $m$ times and Case 2 occurs the remaining $m(m-1)$ times. Therefore, we have

$$\mathbb{E}\|\widehat{B} - B\|_{\mathrm{HS}}^2 = \frac{Q_2}{m} + \frac{m-1}{m}\frac{1}{n^2}\sum_{i_j,i_h=1}^{n} Z_{i_j,i_h}^{j,h} \tag{136}$$

Now, for the second term on the right hand side, Case 2.1 occurs $n$ times while Case 2.2 occurs the remaining $n(n-1)$ times, leading to the desired result. □

**Lemma 18.** *With the notation of Lemma 17, we have*

$$\mathrm{Tr}(\mathfrak{B}) = \mathfrak{b}_1 - \mathfrak{b}_2 \tag{137}$$

*where*

$$\mathfrak{b}_1 = \int \langle g^*(x,p), g^*(x,p')\rangle_{\mathcal{H}} \, k\big((x,p),(x,p')\big) \, d\pi(p|x) d\pi(p'|x) d\rho_X(x) \tag{138}$$

$$\mathfrak{b}_2 = \int \langle g^*(x,p), g^*(x',p')\rangle_{\mathcal{H}} \, k\big((x,p),(x',p')\big) \, d\pi(p|x) d\pi(p'|x') d\rho_X(x) d\rho_X(x'). \tag{139}$$

*Proof.* Note that by definition of $\eta$ and the reproducing property of the kernel $k$, for any $x, x' \in X$, $p, p' \in P$ and $w, w' \in [Y]$ the following holds

$$\mathrm{Tr}(\eta_{x,p,w}^* \eta_{x',p',w'}) = \langle \varphi(w), \varphi(w')\rangle_{\mathcal{H}} \, k\big((x,p),(x',p')\big) - \mathrm{Tr}\left(B^*\left(k_{x,p} \otimes \varphi(w)\right)\right) \tag{140}$$

$$- \mathrm{Tr}\left(B^*\left(k_{x',p'} \otimes \varphi(w')\right)\right) + \mathrm{Tr}(B^*B). \tag{141}$$

Then, by definition of $B = \mathbb{E}\, k_{x,p} \otimes \varphi(w)$, we have

$$\mathrm{Tr}(\mathfrak{B}) = \int \mathrm{Tr}\left(\eta_{x,p,w}^* \eta_{x,p',w'}\right) \, d\mu(w|y, x, p) d\mu(w'|y, x, p') d\pi(p|x) d\pi(p'|x) d\rho(y, x) \tag{142}$$

$$= -\mathrm{Tr}(B^*B) + \int \langle \varphi(w), \varphi(w')\rangle_{\mathcal{H}} \, k\big((x,p),(x',p')\big) \, d\mu(w|y, x, p) d\mu(w'|y, x, p') \times \tag{143}$$

$$\times \, d\pi(p|x) d\pi(p'|x) d\rho(y, x) \tag{144}$$

$$= -\mathrm{Tr}(B^*B) + \int \langle g^*(x,p), g^*(x,p')\rangle_{\mathcal{H}} \, k\big((x,p),(x,p')\big) \, d\pi(p|x) d\pi(p'|x) d\rho_X(x) \tag{145}$$

$$= \mathfrak{b}_1 - \mathrm{Tr}(B^*B), \tag{146}$$

where in the third equality we used the definition of $g^*(x,p) = \int \varphi(w) \, d\mu(w|y, x, p) d\rho(y|x)$. Moreover, since $B$ can be written in terms of $g^*$ as

$$B = \int k_{x,p} \otimes g^*(x,p) \, d\pi(p|x) d\rho_X(x) \tag{147}$$

we have

$$\mathrm{Tr}(B^*B) = \int \langle g^*(x,p), g^*(x',p')\rangle_{\mathcal{H}} \ k\big((x,p),(x',p')\big) \ d\pi(p|x)d\pi(p'|x')d\rho_X(x)d\rho_X(x')$$

(148)

$$= \mathfrak{b}_2,$$

(149)

as desired. $\qquad\square$

### F.3  Learning bound in expectation

We introduce here the assumption that the target function $g^*$ of the learning problem belongs to the RKHS where we are performing the optimization.

**Assumption 3.** *There exists a $\mathsf{G} \in \mathcal{H} \otimes \mathcal{F}$, such that almost everywhere on $X \times P$,*

$$\mathsf{G}k_{x,p} = g^*(x,p).$$

The following results will leverage the assumption above.

**Lemma 19.** *Under Assumption 3,*

$$\mathrm{Tr}(\mathfrak{B}) \ \leqslant \ \|\mathsf{G}\|^2\mathrm{Tr}(\mathfrak{C}),$$

(150)

*Proof.* We begin first observing that $\mathfrak{C}$ is positive semidefinite since

$$\mathfrak{C} = \int \zeta_{x,p}\zeta_{x,p'} \ d\pi(p|x)d\pi(p'|x)d\rho_X(x) = \mathbb{E} \ \zeta_x\zeta_x$$

(151)

is the expectation of the random variable $\zeta_x\zeta_x$, where $\zeta_x = \int \zeta_{x,p} \ d\pi(p|x)$ is positive semidefinite. Moreover, by the definition of $\mathfrak{C}$ in terms of $\zeta_{x,p} = k_{x,p} \otimes k_{x,p} - C$, we have

$$\mathfrak{C} = \int \Big( k_{x,p} \otimes k_{x,p'} \Big) \big((x,p),(x,p')\big) - \Big( k_{x,p} \otimes k_{x,p} \Big)C \ d\pi(p|x)\pi(p'|x)\rho_X(x)$$

(152)

$$+ \int C^2 - C\Big( k_{x,p'} \otimes k_{x,p'} \Big) \ d\pi(p|x)\pi(p'|x)\rho_X(x)$$

(153)

$$= -C^2 + \int \Big( k_{x,p} \otimes k_{x,p'} \Big) \big((x,p),(x,p')\big) \ d\pi(p|x)\pi(p'|x)\rho_X(x)$$

(154)

where we have used the definition of $C = \mathbb{E} \ k_{x,p} \otimes k_{x,p}$.

Now note that under Assumption 3, for any $x, x' \in X$ and $p, p' \in P$

$$\langle g^*(x,p), g^*(x',p')\rangle_{\mathcal{H}} = \langle \mathsf{G}k_{x,p}, \mathsf{G}k_{x',p'}\rangle_{\mathcal{H}} = \mathrm{Tr}\Big( G^*G \Big( k_{x,p} \otimes k_{x',p'} \Big)\Big).$$

(155)

Therefore, substituting the above equation in $\mathfrak{b}_1$ and $\mathfrak{b}_2$ defined in Lemma 18, we have

$$\mathrm{Tr}(\mathfrak{B}) = \mathfrak{b}_1 - \mathfrak{b}_2$$

(156)

$$= \mathrm{Tr}\Big( G^*G \Big[\int \Big( k_{x,p} \otimes k_{x,p'} \Big)\big((x,p),(x,p')\big) \ d\pi(p|x)\pi(p'|x)\rho_X(x) - C^2\Big]\Big)$$

(157)

$$= \mathrm{Tr}(G^*G \ \mathfrak{C})$$

(158)

$$\leqslant \|G\|^2\mathrm{Tr}(\mathfrak{C})$$

(159)

where the last inequality follows from the fact that both $G^*G$ and $\mathfrak{C}$ are positive semidefinite. $\qquad\square$

**Theorem 20.**

$$\mathbb{E}\,\mathcal{E}(\widehat{f}) - \mathcal{E}(f^*) \ \leqslant \ 2\,\mathsf{c}_\triangle\mathsf{g}\left[\lambda^{1/2} + 2\sqrt{2}\left(1 + \left(\frac{\mathsf{r}^2}{\lambda^2m} + \frac{\mathsf{q}}{\lambda^2n}\right)^{1/2}\right)^{1/2}\left(\frac{\mathsf{r}^2}{\lambda m} + \frac{\mathsf{q}}{\lambda n}\right)^{1/2}\right].$$

*In particular when $\lambda \geq \sqrt{\frac{\mathsf{r}^2}{m} + \frac{\mathsf{q}}{n}}$, then*

$$\mathbb{E}\,\mathcal{E}(\widehat{f}) - \mathcal{E}(f^*) \ \leqslant \ 12\,\mathsf{c}_\triangle\mathsf{g}\left(\frac{\mathsf{r}^2}{\lambda m} + \frac{\mathsf{q}}{\lambda n} + \lambda\right)^{1/2}.$$

*Proof.* By the comparison inequality in Thm. 9, we have that

$$\mathbb{E}\,\mathcal{E}(\widehat{f}\,) - \mathcal{E}(f^*) \leqslant 2\mathsf{c}_\triangle\,\mathbb{E}\|\widehat{g} - g^*\|_{L^2(X\times P, \pi\rho_X, \mathcal{H})}.$$

To bound $\mathbb{E}\|\widehat{g} - g^*\|_{L^2(X\times P, \pi\rho_X, \mathcal{H})}$ we need to control some auxiliary quantities. With the notation of Thm. 11 and Lemmas 14, 17 and 19, we have

$$\mathbb{E}\beta_1^2 \leqslant \frac{Q_1}{m} + \frac{\mathrm{Tr}(\mathfrak{C})}{n} =: V, \quad \mathbb{E}\beta_2^2 \leqslant \|\mathsf{G}\|V.$$

In particular note that $\mathrm{Tr}(\mathfrak{C}) = \mathsf{q}$, by Lemma 15 and that by definition of $Q_1$, $\mathsf{r}$ and $C$ we have

$$Q_1 := \mathbb{E}k_{x,p} \otimes k_{x,p} - C_{\mathrm{HS}}^2 \tag{160}$$

$$= \mathrm{Tr}\left(\mathbb{E}\left((x,p),(x,p)\right)(k_{x,p} \otimes k_{x,p}) - 2C(k_{x,p} \otimes k_{x,p}) + C^2\right) \tag{161}$$

$$= \mathrm{Tr}\left(\mathbb{E}\left((x,p),(x,p)\right)(k_{x,p} \otimes k_{x,p}) - C^2\right) \leqslant \mathsf{r}\mathrm{Tr}\left(\mathbb{E}\left(k_{x,p} \otimes k_{x,p}\right)\right) \leqslant \mathsf{r}^2. \tag{162}$$

Moreover, by Assumption 3 we have that $G = S\mathsf{G}$ and so

$$\mathcal{A}_{1/2}(\lambda) = \|L_\lambda^{-1/2}G\|_{\mathrm{HS}(\mathcal{H}, L^2)} = \|L_\lambda^{-1/2}S\mathsf{G}\|_{\mathrm{HS}(\mathcal{H}, L^2)} \leqslant \|L_\lambda^{-1/2}S\|\|\mathsf{G}\|_{\mathrm{HS}(\mathcal{H}, \mathcal{F})} \leqslant \|\mathsf{G}\|_{\mathrm{HS}(\mathcal{H}, \mathcal{F})}.$$

Analogously

$$\mathcal{A}_1(\lambda) = \|L_\lambda^{-1}G\|_{\mathrm{HS}(\mathcal{H}, L^2)} \leqslant \|L_\lambda^{-1/2}\|\|L_\lambda^{-1/2}G\|_{\mathrm{HS}(\mathcal{H}, L^2)} = \lambda^{-1/2}\mathcal{A}_{1/2}(\lambda) \leqslant \lambda^{-1/2}\|\mathsf{G}\|_{\mathrm{HS}(\mathcal{H}, \mathcal{F})}.$$

By plugging the bounds above in the result of Thm. 12, we have

$$\mathbb{E}\|\widehat{g} - g^*\|_{L^2(X\times P, \pi\rho_X, \mathcal{H})} \leqslant 2\sqrt{2}\|\mathsf{G}\|_{\mathrm{HS}(\mathcal{H}, \mathcal{F})}\sqrt{1 + \frac{V^{1/2}}{\lambda}}\sqrt{\frac{V}{\lambda}} + \|\mathsf{G}\|_{\mathrm{HS}(\mathcal{H}, \mathcal{F})}\lambda^{1/2}.$$

By selecting $\lambda \geq V^{1/2}$, we have

$$\mathbb{E}\|\widehat{g} - g^*\|_{L^2(X\times P, \pi\rho_X, \mathcal{H})} \leqslant 4\|\mathsf{G}\|_{\mathrm{HS}(\mathcal{H}, \mathcal{F})}\sqrt{\frac{V}{\lambda}} + \|\mathsf{G}\|_{\mathrm{HS}(\mathcal{H}, \mathcal{F})}\lambda^{1/2} \tag{163}$$

$$\leqslant 4\|\mathsf{G}\|_{\mathrm{HS}(\mathcal{H}, \mathcal{F})}\left(\sqrt{\frac{V}{\lambda}} + \lambda^{1/2}\right) \tag{164}$$

$$\leqslant 4\sqrt{2}\|\mathsf{G}\|_{\mathrm{HS}(\mathcal{H}, \mathcal{F})}\left(\frac{V}{\lambda} + \lambda\right)^{1/2}, \tag{165}$$

since $a^{1/2} + b^{1/2} \leqslant \sqrt{2(a+b)}$ for any $a, b > 0$. $\qquad\square$

We conclude with a corollary of Thm. 20 that frames the result within the notation and setting of the main paper and which will be useful to prove Thm. 4.

In particular, in the following we will consider the standard assumption in the context of non-parametric estimation [7] that $g^* \in \mathcal{G} = \mathcal{H} \otimes \mathcal{F}$, where $\mathcal{F}$ is the reproducing kernel Hilbert space [3] associated to the kernel in (10). The learning rates of $\widehat{f}$ will depend on the following four constants $\mathsf{g}, \mathsf{r}, \mathsf{c}_\triangle, \mathsf{q}$, where

$$\mathsf{g} = \|g^*\|_{\mathcal{G}}, \qquad \mathsf{r} = \sup_{x\in X, p\in P} k((x,p),(x,p)), \qquad \mathsf{c}_\triangle^2 = \sup_{z\in Z, x\in X} \mathbb{E}_{p|x}\|\psi(z,x,p)\|_{\mathcal{H}}^2. \tag{166}$$

Note that the quantities above are rather natural: $\mathsf{r}$ is an upper bound on the kernel $k$, $\mathsf{c}_\triangle$ measures the "complexity" of the loss $\triangle$ and $\mathsf{g}$ quantifies the regularity of $\rho$ in terms of the hypothesis space $\mathcal{F}$ associated to $k$. We will see in Lemma 3 that the latter is related to between-locality. Finally,

$$\mathsf{q} = \mathbb{E}_{x,x'}\mathbb{E}_{p,q|x,r|x'}\,\mathsf{C}_{p,q}(x,x') \qquad \mathsf{C}_{p,q}(x,x') = \left[k((x,p),(x,q))^2 - k((x,p),(x',r))^2\right] \tag{167}$$

where $\mathbb{E}_{p,q|x}[\cdot]$ is a shorthand for $\sum_{p,q\in P}\pi(p|x)\pi(q|x)[\cdot]$ (analogously for $\mathbb{E}_{r|x}$). This quantity will be key to capture and leverage the within-locality assumption. In particular, it will allow us to quantify explicitly the advantages of using our locality-aware estimator.

**Theorem 5.** *Let $\widehat{f}$ as in (7) with i.i.d. training set and auxiliary dataset sampled according to Alg. 1. Let $\triangle$ be SELF, $Z$ compact, $g^* \in \mathcal{G}$ and $\lambda \geq (\mathsf{r}^2/m + \mathsf{q}/n)^{1/2}$. Then*

$$\mathbb{E}\left[\mathcal{E}(\widehat{f}) - \mathcal{E}(f^*)\right] \leqslant 12\, \mathsf{c}_\triangle\, \mathsf{g}\left(\frac{\mathsf{r}^2}{\lambda m} + \frac{\mathsf{q}}{\lambda n} + \lambda\right)^{1/2}. \qquad (20)$$

*Proof.* The desired result corresponds to the second statement of Theorem 20. $\qquad\square$

Thm. 5 characterizes the learning rates of $\widehat{f}$ under standard regularity assumption on the problem without relying on locality assumptions. In particular, we note that when $m \propto n$ and $\lambda \propto n^{-1/2}$, the bound recovers the excess risk bounds of structure prediction *without parts* [8, 9] of order $O(n^{-1/4})$.

# G   Learning Rates with the effect of parts

In this section we prove Thm. 4, studying the effect of between-locality and within-locality on the learning problem. In particular, we consider here the natural generalization of between-locality Assumption 1 to the case where the parts of $y$ are sampled non-deterministically from $\mu$.

**Assumption 4.** *There exist two spaces $[X]$ and $[Y]$ of parts on $X$ and $Y$ respectively and a conditional probability distribution $\bar{\mu}$ on $[Y]$ with respect to $[X]$, such that*

$$\bar{\mu}(w|x_p) = \int \mu(w|y,x,p)d\rho(y|x) \qquad (168)$$

Clearly, Assumption 4 formalizes the concept of between-locality and recovers it when $\mu$ corresponds to

$$\mu(\cdot|y,x,p) = \delta_{y_p}(\cdot) \qquad (169)$$

where $\delta$ denotes the Dirac's delta on the point $y_p \in [Y]$. Indeed, in this case we are requiring $w = y_p$ to depend exclusively on $x_p$ for any $p \in P$, hence to be conditionally independent with respect to $x$. Moreover, we are requiring such distribution $\bar{\mu}$ to be the same for any $p \in P$, hence recovering Assumption 1. The following result is therefore a generalization of Lemma 3, which is recovered as a corollary.

**Lemma 21.** *Under Assumption 4, $g^*$ is such that $g^*(x,p) = \bar{g}^*(x_p)$ for any $x \in X$ and $p \in P$, where $\bar{g}^* : [X] \to \mathcal{H}$ is such that*

$$\bar{g}^*(\xi) = \int \varphi(w)\, d\bar{\mu}(w|\xi) \qquad (170)$$

*almost surely on $[X]$.*

*Proof.* The result follows directly from Assumption 4 and the definition of $g^*$

$$g^*(x,p) = \int \varphi(w)\, d\mu(w|y,x,p)d\rho(y|x) = \int \varphi(w)\, d\bar{\mu}(w|x_p) = \bar{g}^*(x_p), \qquad (171)$$

as desired. $\qquad\square$

**Assumption 5.** *Denote by $\bar{k} : [X] \times [X] \to \mathbb{R}$ the reproducing kernel on $[X]$ with associated rkhs denoted by $\overline{\mathcal{G}}$, defined as for all $x, x' \in X$ and $p, p' \in P$*

$$\big((x,p),(x',p')\big) = \bar{k}(x_p, x'_{p'}) \qquad (172)$$

**Assumption 6.** *There exists $A_0 \in \mathcal{H} \otimes \overline{\mathcal{G}}$ such that the function $\bar{g}^* : [X] \to \mathcal{H}$ can be written as*

$$\bar{g}^*(\eta) = A_0 \bar{k}_\eta.$$

**Lemma 22.** *Under Assumption 5, we have that $\mathcal{F} = \{g \circ \mathsf{i}_X \mid g \in \overline{\mathcal{G}}\}$, with inner product $\langle g \circ \mathsf{i}_X, g' \circ \mathsf{i}_X \rangle_{\mathcal{F}} = \langle g, g' \rangle_{\overline{\mathcal{G}}}$ is a reproducing kernel Hilbert space on $X \times P$, with kernel $k((x,p),(x',p')) = \bar{k}(x_p, x'_{p'})$. Moreover there exists a linear unitary operator $U : \overline{\mathcal{G}} \to \mathcal{F}$ such that $Ug = g \circ \mathsf{i}_X \in \mathcal{F}$ for any $g \in \overline{\mathcal{G}}$.*

*In particular under Assumptions 4 to 6, we have that Assumption 3 is satisfied for $\mathsf{G} = A_0 U^*$, and*

$$\|g^*\|_{\mathcal{H} \otimes \mathcal{F}} := \|\mathsf{G}\|_{\mathrm{HS}(\mathcal{F},\mathcal{H})} = \|A_0\|_{\mathrm{HS}(\bar{\mathcal{G}},\mathcal{H})} = \|\bar{g}^*\|_{\mathcal{H} \otimes \bar{\mathcal{G}}}.$$

*Proof.* By definition $\overline{\mathcal{G}}$ is the RKHS associated to the kernel $\bar{k}$ on $[X]$, where the scalar product $\langle \cdot, \cdot \rangle_{\overline{\mathcal{G}}}$ is defined such that $\langle \bar{k}_\eta, \bar{k}_\zeta \rangle_{\overline{\mathcal{G}}} = \bar{k}(\eta, \zeta)$, for any $\eta, \zeta \in [X]$ and $\overline{\mathcal{G}}$ is the closure of $\overline{\mathcal{G}}_0 = \mathrm{span}\{\bar{k}(\eta, \cdot) \mid \eta \in [X]\}$ w.r.t. $\langle \cdot, \cdot \rangle_{\overline{\mathcal{G}}}$. Similarly $\mathcal{F}$ is the RKHS associated to the kernel $k$ such that the scalar product $\langle \cdot, \cdot \rangle_{\mathcal{F}}$ is defined as $\langle k_{x,p}, k_{x',p'} \rangle_{\mathcal{F}} = \bar{k}(\mathrm{i}_X(x,p), \mathrm{i}_X(x',p'))$, for all $(x,p),(x',p') \in X \times P$. Note that by definition of $\mathcal{F}$, we have that $\mathcal{F}$ is the closure of $\mathcal{F}_0$ w.r.t. $\langle \cdot, \cdot \rangle_{\mathcal{F}}$, with

$$\mathcal{F}_0 = \mathrm{span}\{k((x,p),(\cdot,\cdot)) \mid (x,p) \in X \times P\} \tag{173}$$

$$= \mathrm{span}\{\bar{k}(\mathrm{i}_X(x,p), \mathrm{i}_X(\cdot,\cdot)) \mid (x,p) \in X \times P\} \tag{174}$$

$$= \mathrm{span}\{\bar{k}(\eta, \mathrm{i}_X(\cdot,\cdot)) \mid \eta \in [X]\} \tag{175}$$

$$= \overline{\mathcal{G}}_0 \circ \mathrm{i}_X. \tag{176}$$

Now, since for any $\eta, \zeta \in [X]$ there exist $(x,p),(x',p') \in [X]$ such that $\eta = \mathrm{i}_X(x,p), \zeta = \mathrm{i}_X(x',p')$, we have that,

$$\langle \bar{k}(\eta, \mathrm{i}_X(\cdot,\cdot)), \bar{k}(\zeta, \mathrm{i}_X(\cdot,\cdot)) \rangle_{\mathcal{F}} = \langle \bar{k}(\mathrm{i}_X(x,p), \mathrm{i}_X(\cdot,\cdot)), \bar{k}(\mathrm{i}_X(x',p'), \mathrm{i}_X(\cdot,\cdot)) \rangle_{\mathcal{F}} \tag{177}$$

$$= \bar{k}(\mathrm{i}_X(x,p), \mathrm{i}_X(x',p')) = \bar{k}(\eta, \zeta) = \langle \bar{k}_\eta, \bar{k}_\zeta \rangle_{\overline{\mathcal{G}}}. \tag{178}$$

So, let $f, f' \in \mathcal{F}_0$, by definition we have $f = g \circ \mathrm{i}_X$ and $f' = g' \circ \mathrm{i}_X$ with $g, g' \in \overline{\mathcal{G}}_0$. Moreover by definition of $g, g'$ there exist $n, m \in \mathbb{N}$ and $\eta_1, \ldots, \eta_n, \zeta_1, \ldots, \zeta_m \in [X]$ and $\alpha_1, \ldots, \alpha_n, \beta_1, \ldots, \beta_m \in \mathbb{R}$ such that $g(\cdot) = \sum_{i=1}^n \alpha_i \bar{k}(\eta_i, \cdot)$ and analogously $g'(\cdot) = \sum_{j=1}^m \beta_j \bar{k}(\zeta_j, , \cdot)$.

Now we show that $\langle g \circ \mathrm{i}_X, g' \circ \mathrm{i}_X \rangle_{\mathcal{F}} = \langle g, g' \rangle_{\overline{\mathcal{G}}}$ for $g, g' \in \overline{\mathcal{G}}_0$ and then we extend it to $\overline{\mathcal{G}}$. First we recall that the composition on the right is linear, indeed

$$(\alpha f + \beta g) \circ h = \alpha(f \circ h) + \beta(g \circ h),$$

for any $\alpha, \beta \in \mathbb{R}$, any function $f, g : A \to \mathbb{R}$ and $h : B \to A$, and $A, B$ two sets. Then we have

$$\langle f, f' \rangle_{\mathcal{F}} = \langle g \circ \mathrm{i}_X, g' \circ \mathrm{i}_X \rangle_{\mathcal{F}} = \left\langle \left( \sum_{i=1}^n \alpha_i \bar{k}(\eta_i, \cdot) \right) \circ \mathrm{i}_X, \left( \sum_{j=1}^m \beta_j \bar{k}(\zeta_j, \cdot) \right) \circ \mathrm{i}_X \right\rangle \tag{179}$$

$$= \left\langle \sum_{i=1}^n \alpha_i \bar{k}(\eta_i, \mathrm{i}_X(\cdot,\cdot)), \sum_{j=1}^m \beta_j \bar{k}(\zeta_j, \mathrm{i}_X(\cdot,\cdot)) \right\rangle \tag{180}$$

$$= \sum_{i=1}^n \sum_{j=1}^m \alpha_i \beta_j \left\langle \bar{k}(\eta_i, \mathrm{i}_X(\cdot,\cdot)), \bar{k}(\zeta_j, \mathrm{i}_X(\cdot,\cdot)) \right\rangle_{\mathcal{F}} \tag{181}$$

$$= \sum_{i=1}^n \sum_{j=1}^m \alpha_i \beta_j \left\langle \bar{k}_{\eta_i}, \bar{k}_{\zeta_j} \right\rangle_{\overline{\mathcal{G}}} = \left\langle \sum_{i=1}^n \alpha_i \bar{k}_{\eta_i}, \sum_{j=1}^m \beta_j \bar{k}_{\zeta_j} \right\rangle_{\overline{\mathcal{G}}} \tag{182}$$

$$= \langle g, g' \rangle_{\overline{\mathcal{G}}}. \tag{183}$$

By noting that

$$\|g_n \circ \mathrm{i}_X - g_m \circ \mathrm{i}_X\|_{\mathcal{F}} = \|(g_n - g_m) \circ \mathrm{i}_X\|_{\mathcal{F}} = \|g_n - g_m\|_{\overline{\mathcal{G}}}$$

for any Cauchy sequence $(g_n)_{n \in \mathbb{N}}$ in $\overline{\mathcal{G}}_0$, and the fact that $\mathcal{F}_0 = \overline{\mathcal{G}} \circ \mathrm{i}_X$ and that $\langle g \circ \mathrm{i}_X, g \circ \mathrm{i}_X \rangle_{\mathcal{F}} = \langle g, g' \rangle_{\overline{\mathcal{G}}}$, for $g, g' \in \overline{\mathcal{G}}_0$, then we have that $\mathcal{F} = \overline{\mathcal{G}} \circ \mathrm{i}_X$, and that $\langle g \circ \mathrm{i}_X, g \circ \mathrm{i}_X \rangle_{\mathcal{F}} = \langle g, g' \rangle_{\overline{\mathcal{G}}}$, for $g, g' \in \overline{\mathcal{G}}$.

Now denote by $U : \overline{\mathcal{G}} \to \mathcal{F}$ the operator such that $Ug = g \circ \mathrm{i}_X$. First note that $U$ is linear, indeed

$$U(\alpha g + \beta h) = (\alpha g + \beta h) \circ \mathrm{i}_X = \alpha(g \circ \mathrm{i}_X) + \beta(h \circ \mathrm{i}_X) = \alpha Ug + \beta Uh,$$

for any $g, h \in \overline{\mathcal{G}}$ and $\alpha, \beta \in \mathbb{R}$. Moreover we show that $U$ is a partial isometry, indeed

$$\|Ug\|_{\mathcal{F}}^2 = \|g \circ \mathrm{i}_X\|_G^2 = \langle g \circ \mathrm{i}_X, g \circ \mathrm{i}_X \rangle_{\mathcal{F}} = \langle g, g \rangle_{\overline{\mathcal{G}}} = \|g\|_{\overline{\mathcal{G}}}^2.$$

Finally by applying the result above to $g^*$ and $\bar{g}^*$, under Assumptions 4 to 6, we have that $\mathsf{G} = A_0 U^*$ and so, by using the isomorphism between $\mathcal{H} \otimes \mathcal{F}$ and $\mathrm{HS}(\mathcal{F}, \mathcal{H})$, we have

$$\|g^*\|_{\mathcal{H} \otimes \mathcal{F}} := \|\mathsf{G}\|_{\mathrm{HS}(\mathcal{F}, \mathcal{H})} = \|A_0\|_{\mathrm{HS}(\overline{\mathcal{G}}, \mathcal{H})} = \|\bar{g}^*\|_{\mathcal{H} \otimes \overline{\mathcal{G}}},$$

as desired. $\qquad\square$

**Assumption 7.** *The distribution $\pi(\cdot|x) = \pi(\cdot|x')$ for any $x, x' \in X$. For the sake of simplicity we will denote it by $\pi(\cdot)$.*

**Lemma 23.** *Under Assumption 7, the following hold*

$$\mathsf{q} \;=\; \mathbb{E}_{pq}\, \mathfrak{c}_{pq}, \tag{184}$$

*where, for $p, q \in P$*

$$\mathfrak{c}_{pq} \;=\; \mathbb{E}_{x,x'}\left[ k((x,p),(x,q))^2 - k((x,p),(x',q))^2 \right]. \tag{185}$$

*Proof.* First note that with the definitions of Lemma 15, we have

$$\mathsf{q} = \mathfrak{c}_1 - \mathfrak{c}_2$$

by Lemma 15 . Under Assumption 7 we can denote $\pi(\cdot|x) = \pi(\cdot)$ without ambiguity. Then with the notation of Lemma 15, we have

$$\mathfrak{c}_1 = \int k\big((x,p),(x,q)\big)^2 \, d\pi(p)d\pi(q)d\rho_X(x) \tag{186}$$

$$= \mathbb{E}_{p,q} \int k\big((x,p),(x,q)\big)^2 \, d\rho_X(x) \tag{187}$$

$$= \mathbb{E}_{p,q}\mathbb{E}_x k\big((x,p),(x,q)\big)^2 \tag{188}$$

Analogously for $\mathfrak{c}_2$

$$\mathfrak{c}_2 = \int k\big((x,p),(x',q)\big)^2 \, d\pi(p)d\pi(q) \, d\rho_X(x)\rho_X(x') \tag{189}$$

$$= \mathbb{E}_{p,q} \int k\big((x,p),(x,q)\big)^2 \, d\rho_X(x)\rho_X(x') \tag{190}$$

$$= \mathbb{E}_{p,q}\mathbb{E}_{x,x'} k\big((x,p),(x,q)\big)^2 \tag{191}$$

$$\square$$

As an immediate corollary in the case where $P$ has finite cardinality, we have

**Corollary 24.** *Under the same assumptions of Thm. 5, let $k$ denote the restriction kernel defined in (15) in terms of $\bar{k} : [X] \times [X] \to \mathbb{R}$. Let $\pi(p|x) = \frac{1}{|P|}$ for any $x \in X$ and $p \in P$. Then, the constant $\mathsf{q}$ in (167) can be factorized as*

$$\mathsf{q} = \frac{1}{|P|^2} \sum_{p,q \in P} \mathsf{C}_{p,q},$$
$$\mathsf{C}_{p,q} = \mathbb{E}_{x,x'}\left[ \bar{k}(x_p, x_q)^2 - \bar{k}(x_p, x'_q)^2 \right]. \tag{192}$$

### G.1   Proof of Theorem 4

*Proof.* This proof consists in applying Theorem 5 with $\lambda = \sqrt{\mathsf{r}^2/m + \mathsf{q}/n}$, and taking into account between-locality and within-locality.

First, under the between-locality condition formalized in our measure theoretic setting as Assumption 4, there exists a $\bar{g}^* : [X] \to \mathcal{H}$ such that $g^*(x,p) = \bar{g}^*(x_p)$ for any $x \in X$ and $p \in P$ as proven by Lemma 21. So the restriction kernel can learn $\bar{g}^*$ if it is rich enough, that is $\bar{g}^* \in \mathcal{H} \otimes \bar{\mathcal{F}}$ (here formalized as Assumption 6, with $\bar{\mathcal{F}}$ denoted by $\bar{\mathcal{G}}$). Then we can apply Lemma 22, that guarantees the applicability of Theorem 5.

Second, by the assumption on the fact that $\pi(p|x) = 1/|P|$, we can apply Cor. 24 and then the within-locality condition of Assumption 2, obtaining the desired result. $\square$

# H  Universal Consistency

A natural question is how to design a structured prediction estimator that is both able to leverage the locality assumptions, when they hold, and be universally consistent even when there is no locality. The following remark addresses this questions and concludes our theoretical analysis.

**Theorem 25** (Universal Consistency). *Let $\triangle$ be SELF and $Z$ a compact set. Let $k$ be a bounded continuous universal kernel on $X \times P$. Let $\widehat{f}_n$ as in (7) with i.i.d. training set and auxiliary dataset sampled according to Sec. 4, with $m \propto n$. Then*

$$\lim_{n \to \infty} \mathcal{E}(\widehat{f}_n) = \inf_{f : X \to Z} \mathcal{E}(f) \quad \text{with probability } 1. \tag{193}$$

*Proof.* Appendix H.1 is devoted to the proof. $\qquad\square$

The requirement of universality for the kernel is a standard assumption (see [38]). An example of continuous universal kernel on $X \times P$ is $k((x,p),(x',p')) = k_0(x,x')\,\delta_{p,p'}$ where $k_0$ is any unversal kernel on $X$, e.g. the Gaussian $k_0(x,x') = \exp(-\|x-x'\|^2)$.

While the proposed estimator is consistent with the kernel described above, it is not able to benefit from the effect of locality. In the following we comment on how to obtain a kernel that guarantees both universal consistency while leveraging locality at the same time.

**Remark 6** (Universal and Local Kernels). *By construction, the restriction kernel allows to learn only functions $g^* : X \times P \to \mathcal{H}$ such that $g^*(x,p) = \bar{g}^*(x_p)$. Consequently, the corresponding structured prediction estimator is not universal. However, in Thm. 4 we have observed that under the locality assumptions, the restriction kernel achieves significantly faster rates with respect to universal kernels that are not tailored to account for the part structure on the input.*

*Interestingly, it is possible to design a kernel able to take the best of both worlds, leading to an estimator that is universal but also able to leverage the parts-based structure of a learning problem when possible. We obtain this kernel as the sum $k_B = k_U + k_L$ of a universal kernel $k_U$ on $X \times P$ and a restriction (or "local") kernel $k_L$. Indeed, as shown in Appendix I.4, the kernel $k_B$ is universal, hence Thm. 25 applies to the corresponding estimator $\widehat{f}$. Moreover, under the locality assumptions, a result identical to Thm. 4 holds for the estimator trained with $k_B$.*

## H.1  Proof of Thm. 25

The proof is exactly the same as in Theorem 4 Section B.3 of the supplementary materials of [8], where instead of using their comparison inequality (their Thm. 2) we use our Thm. 9 and instead using their Lemma 18 we use our Theorem 29 that is proven at the end of this section. First we introduce some concentration inequalities for separable Hilbert spaces.

**Proposition 26.** *Let $\delta \in (0,1]$ and $m \in \mathbb{N}$. Let $\mathcal{H}$ be a separable Hilbert space. Let $\zeta_1, \ldots, \zeta_m$ be independently distributed $\mathcal{H}$-valued random variables. Let $R > 0$ be such that* ess sup $\|\zeta_j\|_{\mathcal{H}} \leqslant R$ *for every $j = 1, \ldots, m$. Then,*

$$\Big\| \frac{1}{m} \sum_{j=1}^{m} \big[ \zeta_j - \mathbb{E}\,\zeta_j \big] \Big\|_{\mathcal{H}} \leqslant \frac{4R \log \frac{3}{\delta}}{\sqrt{m}} \tag{194}$$

*with probability at least $1 - \delta$.*

*Proof.* By applying Lemma 2 of [35] with constants $\widetilde{M} = R$ and $\sigma^2 = \sup_j \mathbb{E}\|\zeta_j\|^2 \leqslant R^2$, we obtain

$$\Big\| \frac{1}{m} \sum_{j=1}^{m} \big[ \zeta_j - \mathbb{E}\zeta_j \big] \Big\|_{\mathrm{HS}} \leqslant \frac{2R \log \frac{2}{\delta}}{m} + \sqrt{\frac{2R^2 \log \frac{2}{\delta}}{m}} \tag{195}$$

with probability at least $1 - \delta$. Now, $\log \frac{2}{\delta} \leqslant \log \frac{3}{\delta}$ and $\log 3\delta \geq 1$ for any $\delta \in (0,1]$. Then, we can bound the above inequality by

$$\frac{2R \log \frac{2}{\delta}}{m} + \sqrt{\frac{2R^2 \log \frac{2}{\delta}}{m}} \leqslant \frac{4R \log \frac{3}{\delta}}{\sqrt{m}}, \tag{196}$$

as desired. □

**Remark 7** (Pinelis Inequality for Hilbert-Schmidt Operators). *We recall that the space of Hilbert-Schmidt operators between two separable Hilbert spaces is itself a separable Hilbert space with the Hilbert-Schmidt norm. Therefore, Pinelis inequality in Prop. 26 is directly applicable.*

**Lemma 27.** *Let $C$ and $\widehat{C}$ and $\kappa = \sup_{x,p} \|k_{x,p}\|_{\mathcal{F}}$ defined as Appendix B. Let $\delta \in (0,1]$. Then*

$$\|\widehat{C} - C\| \leqslant 4\kappa^2 \left( \frac{1}{\sqrt{m}} + \frac{1}{\sqrt{n}} \right) \log \frac{6}{\delta} \tag{197}$$

*with probability at least $1 - \delta$.*

*Proof.* Given a dataset $(x_i)_{i=1}^n$, we introduce the operator $\widetilde{C} : \mathcal{F} \to \mathcal{F}$ defined as

$$\widetilde{C} = \frac{1}{n} \sum_{i=1}^n \int_P k_{x_i,p} \otimes k_{x_i,p} \, d\pi(p|x_i). \tag{198}$$

and consider the following decomposition

$$\|\widehat{C} - C\| \leqslant \|\widehat{C} - \widetilde{C}\| + \|\widetilde{C} - C\|. \tag{199}$$

Let $\tau = \delta/2$, in the following we separately bound the terms above in probability and then take the intersection bound.

**Bounding $\|\widehat{C} - \widetilde{C}\|$.** For any $j = 1, \ldots, m$ let $\zeta_j = k_{x_{i_j},p_j} \otimes k_{x_{i_j},p_j}$ with $i_j$ and $p_j$ independently sampled respectively from: the uniform distribution on $\{1, \ldots, n\}$ and the conditional probability $\pi(\cdot|x_{i_j})$. Therefore, for any $j = 1, \ldots, m$

$$\widehat{C} = \frac{1}{m} \sum_{j=1}^m \zeta_j, \qquad \widetilde{C} = \mathbb{E}\, \zeta_j = \frac{1}{n} \sum_{i=1}^n \int_P k_{x_i,p} \otimes k_{x_i,p} \, d\pi(p|x_i) \tag{200}$$

and

$$\text{ess sup } \|\zeta_j\|_{\text{HS}} \leqslant \sup_{x \in X, p \in P} \langle k_{x,p}, k_{x,p} \rangle_{\mathcal{F}} \leqslant \sup_{x \in X, p \in P} \|k_{x,p}\|_{\mathcal{F}}^2 \leqslant \kappa^2$$

We apply Pinelis inequality (see Remark 7), leading to

$$\|\widehat{C} - \widetilde{C}\| \leqslant \|\widehat{C} - \widetilde{C}\|_{\text{HS}} = \Big\|\frac{1}{m} \sum_j \big[\zeta_j - \mathbb{E}\zeta_j\big]\Big\|_{\text{HS}} \leqslant \frac{4\kappa^2 \log \frac{3}{\tau}}{\sqrt{m}} \tag{201}$$

with probability at least $1 - \tau$.

**Bounding $\|\widetilde{C} - C\|$.** For $i = 1, \ldots, n$ let $\eta_i = \int_P k_{x_i,p} \otimes k_{x_i,p} \, d\pi(p|x_i)$ with $x_i$ independently sampled from $\rho_X$. Therefore, for every $i = 1, \ldots, n$,

$$\widetilde{C} = \frac{1}{n} \sum_{i=1}^n \eta_i, \qquad C = \mathbb{E}\, \eta_i = \int_{X \times P} k_{x,p} \otimes k_{x,p} \, d\pi(p|x) d\rho_X(x) \tag{202}$$

and

$$\text{ess sup } \|\eta_i\|_{\text{HS}} \leqslant \sup_{x \in X, p \in P} \|k_{x,p}\|_{\mathcal{F}}^2 \leqslant \kappa^2.$$

We apply again Pinelis inequality, obtaining

$$\|\widetilde{C} - C\| \leqslant \|C - \widetilde{C}\|_{\text{HS}} = \Big\|\frac{1}{n} \sum_{i=1}^n \big[\eta_i - \mathbb{E}\eta_i\big]\Big\|_{\text{HS}} \leqslant \frac{4\kappa^2 \log \frac{3}{\tau}}{\sqrt{n}} \tag{203}$$

with probability at least $1 - \tau$.

By taking the intersection bound of the two events above, we obtain

$$\|\widehat{C} - C\|_{\text{HS}} \leqslant \frac{4\kappa^2 \log \frac{3}{\tau}}{\sqrt{m}} + \frac{4\kappa^2 \log \frac{3}{\tau}}{\sqrt{n}} \tag{204}$$

with probability at least $1 - 2\tau$. By recalling $\tau = \frac{\delta}{2}$ we obtain the desired result. □

**Lemma 28.** *Let $B$, $\widehat{B}$, $\kappa = \sup_{x,p} \|k_{x,p}\|_{\mathcal{F}}$ and $q = \sup_w \|\varphi(w)\|_{\mathcal{H}}$ defined as [Appendix B](). Let $\delta \in (0,1]$. Then*

$$\|\widehat{B} - B\|_{\mathrm{HS}} \leqslant 4\kappa q \left( \frac{1}{\sqrt{m}} + \frac{1}{\sqrt{n}} \right) \log \frac{6}{\delta} \tag{205}$$

*with probability at least $1 - \delta$.*

*Proof.* Given $(x_i, y_i)_{i=1}^n$ a dataset, we introduce the operator $\widetilde{B} : \mathcal{H} \to \mathcal{F}$ defined as

$$\widetilde{B} = \frac{1}{n} \sum_{i=1}^n \int_P k_{x_i,p} \otimes \varphi(w) \, d\mu(w|y_i, x_i, p) d\pi(p|x_i). \tag{206}$$

and consider the following decomposition

$$\|\widehat{B} - B\|_{\mathrm{HS}} \leqslant \|\widehat{B} - \widetilde{B}\|_{\mathrm{HS}} + \|\widetilde{B} - B\|_{\mathrm{HS}}. \tag{207}$$

Let $\tau = \delta/2$, in the following we separately bound the terms above in probability and then take the intersection bound.

**Bounding $\|\widehat{B} - \widetilde{B}\|_{\mathrm{HS}}$.** For any $j = 1, \dots, m$ let $\xi_j = k_{x_{i_j}, p_j} \otimes \varphi(w_j)$ with $i_j, p_j$ and $w_j$ independently sampled respectively from: the uniform distribution on $\{1, \dots, n\}$, the conditional probability $\pi(\cdot|x_{i_j})$ and the conditional probability $\mu(\cdot|y_{i_j}, x_{i_j}, p_j)$. Therefore, for any $j = 1, \dots, m$

$$\widehat{B} = \frac{1}{m} \sum_{j=1}^m \xi_j, \qquad \widetilde{B} = \mathbb{E}\, \xi_j = \frac{1}{n} \sum_{i=1}^n \int_{[Y] \times P} k_{x_i,p} \otimes \varphi(w) \, d\mu(w|x_i, y_i, p) d\pi(p|x_i), \tag{208}$$

moreover

$$\mathrm{ess\,sup}\, \|\xi_j\|_{\mathrm{HS}} \leqslant \sup_{x,p,w} \|k_{x,p} \otimes \varphi(w)\|_{\mathrm{HS}} = \sup_{x,p,w} \|k_{x,p}\|_{\mathcal{F}} \|\varphi(w)\|_{\mathcal{H}} \leqslant \kappa q. \tag{209}$$

We apply Pinelis inequality (see [Remark 7]()), leading to

$$\|\widehat{B} - \widetilde{B}\|_{\mathrm{HS}} = \Big\| \frac{1}{m} \sum_{j=1}^m \big[ \xi_j - \mathbb{E}\, \xi_j \big] \Big\|_{\mathrm{HS}} \leqslant \frac{4\kappa q \log \frac{3}{\tau}}{\sqrt{m}} \tag{210}$$

with probability at least $1 - \tau$.

**Bounding $\|B - \widetilde{B}\|_{\mathrm{HS}}$.** For any $i = 1, \dots, n$, let $\nu_i = \int_{[Y] \times P} k_{x_i,p} \otimes \varphi(w) \, d\mu(w|y_i, x_i, p) d\pi(p|x_i)$ with $(x_i, y_i)$ independently sampled from $\rho$. Then, for any $i = 1, \dots, n$

$$\mathbb{E}\, \nu_i = \int_{[Y] \times Y \times X \times P} k_{x,p} \otimes \varphi(w) \, d\mu(w|y_i, x_i, p) d\pi(p|x_i) d\rho(y,x) \tag{211}$$

$$= \int_{X \times P} k_{x,p} \otimes \left[ \int_{[Y] \times Y} \varphi(w) \, d\mu(w|y_i, x_i, p) d\rho(y|x) \right] d\pi(p|x_i) d\rho_X(x) \tag{212}$$

$$= \int_{X \times P} k_{x,p} \otimes g^*(x,p) \, d\pi(p|x_i) d\rho_X(x) \tag{213}$$

$$= B \tag{214}$$

and $\widetilde{B} = \frac{1}{n} \sum_{i=1}^n \nu_i$. Moreover,

$$\mathrm{ess\,sup}\, \|\nu_i\|_{\mathrm{HS}} \leqslant \sup_{x,y} \int_{[Y] \times P} \|k_{x,p} \otimes \varphi(w)\|_{\mathrm{HS}} \, d\mu(w|y, x, p) d\pi(p|x) \tag{215}$$

$$= \sup_{x,y} \int_{[Y] \times P} \|k_{x,p}\|_{\mathcal{F}} \|\varphi(w)\|_{\mathcal{H}} \, d\mu(w|y, x, p) d\pi(p|x) \tag{216}$$

$$\leqslant \kappa q \sup_{x,y} \int_{[Y] \times P} d\mu(w|y, x, p) d\pi(p|x) \tag{217}$$

$$= \kappa q \tag{218}$$

$$\tag{219}$$

Therefore, applying again Pinelis inequality,

$$\|B - \widetilde{B}\|_{\mathrm{HS}} = \left\| \frac{1}{n} \sum_{i=1}^{n} \left[ \nu_i - \mathbb{E}\nu_i \right] \right\|_{\mathrm{HS}} \leqslant \frac{4\kappa q \log \frac{3}{\tau}}{\sqrt{n}} \tag{220}$$

with probability at least $1 - \tau$.

By taking the intersection bound of the two events above, we obtain

$$\|\widehat{B} - B\|_{\mathrm{HS}} \leqslant \frac{4\kappa q \log \frac{3}{\tau}}{\sqrt{m}} + \frac{4\kappa q \log \frac{3}{\tau}}{\sqrt{n}} \tag{221}$$

with probability at least $1 - 2\tau$ as desired. $\qquad\square$

**Theorem 29.** *Let $\delta \in (0,1]$. Let $Q > 0$, $n \in \mathbb{N}$, $c_Q = 1 + 1/\sqrt{Q}$ and $m = Qn$. Then*

$$\|\widehat{g} - g^*\|_{L^2(X \times P, \pi \rho_X, \mathcal{H})} \leqslant \frac{4\kappa^2 c_Q (\|L_\lambda^{-1/2} G\|_{\mathrm{HS}} + \frac{q}{\kappa}) \log \frac{12}{\delta}}{\sqrt{\lambda n}} \left( 1 + 2\kappa \sqrt{\frac{c_Q \log \frac{12}{\delta}}{\lambda \sqrt{n}}} \right) + \lambda \|L_\lambda^{-1} G\|_{\mathrm{HS}} \tag{222}$$

*with probability at least $1 - \delta$.*

*Proof.* In [Thm. 11](#) we have bounded $\|\widehat{g} - g^*\|_{L^2(X \times P, \pi \rho_X, \mathcal{H})}$ in terms of an analytic expression of $\|C - \widehat{C}\|$ and $\|B - \widehat{B}\|_{\mathrm{HS}}$. We bound these two terms with probability $1 - \tau$ with $\tau = \delta/2$ via [Lemma 27](#) and [Lemma 28](#). We further take the intersection bound to obtain the desired result. $\qquad\square$

# I  Equivalence between SELF and SELF by Parts without assumptions

## I.1  SELF without Parts

We begin by briefly recalling the SELF framework in [8]. We will see that this is a special case of the setting proposed in this work for a special choice of the kernel on $X \times P$.

We recall the definition of SELF introduced in [8] and consider the formulation in [9].

**Definition 3.** *A function $\triangle : Z \times Y \to \mathbb{R}$ is a Structure Encoding Loss Function (SELF) if there exist a Hilbert space $\bar{\mathcal{H}}$ and two maps $\bar{\psi} : Z \to \mathcal{H}$ and $\bar{\varphi} : Y \to \mathcal{H}$ such that*

$$\triangle(z,y) = \langle \bar{\psi}(z), \bar{\varphi}(y) \rangle_{\bar{\mathcal{H}}} \tag{223}$$

*for all $z \in Z, y \in Y$.*

Below we show that the definition of SELF by parts introduced in this work is a refinement of the original one. Since the original definition of SELF did not account for the possibility of $\triangle$ do depend also on the input, below we consider only the case $\triangle(z,y|x) = \triangle(z,y)$. In particular we will assume in [Def. 1](#) that $\pi(p|x) = \pi(p|x')$ for any $x, x' \in X, p \in P$ and denote it $\pi(p)$.

**Lemma 30.** *Let $\triangle : Z \times Y \to \mathbb{R}$ satisfy [Def. 1](#) with*

$$\triangle(z,y) = \sum_{p \in P} \ell(z,y|p)\pi(p) = \sum_{p \in P} \langle \psi(z,p), \varphi(y_p) \rangle_{\mathcal{H}} \tag{224}$$

*Then $\triangle$ satisfies the original SELF definition [Def. 3](#), with $\bar{\mathcal{H}} = \mathcal{H} \otimes \mathbb{R}^P$ and maps $\bar{\psi} : Z \to \bar{\mathcal{H}}$ and $\bar{\varphi} : Y \to \bar{\mathcal{H}}$ such that*

$$\bar{\psi}(z) = (\sqrt{\pi(p)}\psi(z,p))_{p \in P} \qquad and \qquad (\sqrt{\pi(p)}\varphi(y_p))_{p \in P} \tag{225}$$

*In particular, we have that the constant $\mathsf{c}_\triangle$ is*

$$\mathsf{c}_\triangle = \sqrt{\sup_{z \in Z} \sum_{p \in P} \pi(p)\|\psi(z,p)\|_{\mathcal{H}}^2} = \sup_{z \in Z} \|\bar{\psi}(z)\|_{\bar{\mathcal{H}}}. \tag{226}$$

*Proof.* Recall that by construction $\bar{\mathcal{H}} = \mathcal{H} \otimes \mathbb{R}^P = \bigoplus_{p \in P} \mathcal{H}$. Therefore, any vector $\eta \in \bar{\mathcal{H}}$ is the collection $(\eta_p)_{p \in P}$ with $\eta_1, \ldots, \eta_P \in \mathcal{H}$ and the corresponding inner product with a $\zeta = (\zeta_p)_{p \in P} \in \bar{\mathcal{H}}$ is

$$\langle \eta, \zeta \rangle_{\bar{\mathcal{H}}} = \sum_{p \in P} \langle \eta_p, \zeta_p \rangle_{\mathcal{H}}. \tag{227}$$

Plugging the definition of $\bar{\psi}$ and $\bar{\varphi}$ in the definition of SELF by parts, we have

$$\triangle(z, y) = \sum_{p \in P} \pi(p) \langle \varphi(z, p), \psi(y_p) \rangle_{\mathcal{H}} \tag{228}$$

$$= \sum_{p \in P} \left\langle \sqrt{\pi(p)} \varphi(z, p), \sqrt{\pi(p)} \psi(y_p) \right\rangle_{\mathcal{H}} \tag{229}$$

$$= \left\langle \bar{\psi}(z), \bar{\varphi}(y) \right\rangle_{\bar{\mathcal{H}}} \tag{230}$$

as required. $\qquad \square$

## I.2 SELF Solution

Given a loss $\triangle$ that is a SELF by parts, we have already observed that the solution $f^* : X \to Z$ of the structured prediction problem in (5), can be characterized in terms of a function $g^* : X \times P \to \mathcal{H}$ introduced in (13). Based on the relation highlighted by Lemma 30, we have the following equivalent characterization

$$f^*(x) = \operatorname*{argmin}_{z \in Z} \ \langle \bar{\psi}(z), h^*(x) \rangle_{\bar{\mathcal{H}}} \tag{231}$$

where now $h^* : X \to \bar{\mathcal{H}}$ is conditional mean embedding of $\bar{\varphi}(y)$ in $\bar{\mathcal{H}}$ with respect to the conditional distribution $\rho(y|x)$. In particular, let $e_p \in \mathbb{R}^P$ denote the $p$-th element of the canonical basis in $\mathbb{R}^P$. Then, for any $\eta \in \mathcal{H}$, $x \in X$ and $p \in P$, we have

$$\langle h^*(x), \eta \otimes e_p \rangle_{\bar{\mathcal{H}}} = \left\langle \int \bar{\varphi}(y) \, d\rho(y|x), \eta \otimes e_p \right\rangle \tag{232}$$

$$= \sqrt{\pi(p)} \left\langle \int \varphi(y_p) \, d\rho(y|x), \eta \right\rangle_{\mathcal{H}} \tag{233}$$

$$= \sqrt{\pi(p)} \left\langle g^*(x, p), \eta \right\rangle_{\mathcal{H}}, \tag{234}$$

and in particular,

$$h^*(x) = (\sqrt{\pi(p)} g^*(x, p))_{p \in P}. \tag{235}$$

We conclude that

$$\|h^*\|^2_{L^2(X, \rho_X, \bar{\mathcal{H}})} = \int \langle h^*(x), h^*(x) \rangle_{\bar{\mathcal{H}}} \ d\rho_X(x) \tag{236}$$

$$= \int \sum_{p \in P} \left\langle \sqrt{\pi(p)} g^*(x, p), \sqrt{\pi(p)} g^*(x, p) \right\rangle_{\mathcal{H}} \ d\rho_X(x) \tag{237}$$

$$= \int \sum_{p \in P} \pi(p) \langle g^*(x, p), g^*(x, p) \rangle_{\mathcal{H}} \ d\rho_X(x) \tag{238}$$

$$= \|g^*\|^2_{L^2 X, \pi \rho_X, \mathcal{H}}. \tag{239}$$

## I.3 If $g^*$ is "simple" (e.g. Assumption 1 holds)

Let $\bar{k}$ be a kernel on $X$ with RKHS $\mathcal{F}$. Let $k$ be a kernel on $X \times P$ defined as $k((x, p), (x', p')) = \bar{k}(x, x') \delta_{p, p'}$, for $x, x' \in X$, $p, p' \in P$. Note that the RKHS associated to $k$ is $\mathcal{F} \otimes \mathbb{R}^P$ with $k_{x,p} = \bar{k}_x \otimes e_p$ and $e_p \in \mathbb{R}^P$ the $p$-th element of the canonical basis of $\mathbb{R}^P$.

**Lemma 31.** *Let* $\mathsf{G} \in \mathcal{H} \otimes \mathcal{F} \otimes \mathbb{R}^P$ *be such that* $g^*(x, p) = \mathsf{G}k_{x,p}$ *for any* $x \in X$ *and* $p \in P$. *Let* $G_1, \ldots, G_P \in \mathcal{H} \otimes \mathcal{F}$ *the operator such that* $G_p \eta = G(\eta \otimes e_p)$ *for any* $p \in P$ *and* $\eta \in \mathcal{F}$. *Then,*

- $\mathsf{G} = \sum_{p \in P} \mathsf{G}_p \otimes e_p.$

- *For any $x \in X$, $h^*(x) = \mathsf{H}\bar{k}_x$ with $\mathsf{H} = \sum_{p \in P} e_p \otimes \sqrt{\pi(p)}\mathsf{G}_p \in \mathbb{R}^P \otimes \mathcal{H} \otimes \mathcal{F}.$*

*In particular*

$$\|\mathsf{G}\|^2_{\mathrm{HS}(\mathcal{F} \otimes \mathbb{R}^P, \mathcal{H})} = \sum_{p \in P} \|\mathsf{G}_p\|^2_{\mathrm{HS}(\mathcal{F}, \mathcal{H})} \qquad \text{and} \qquad \|H\|^2_{\mathrm{HS}(\mathcal{F}, \mathcal{H} \otimes \mathbb{R}^P)} = \sum_{p \in P} \pi(p) \|\mathsf{G}_p\|^2_{\mathrm{HS}(\mathcal{F}, \mathcal{H})}.$$
(240)

**Lemma 32.** *Let $\mathsf{G} \in \mathcal{H} \otimes (\mathcal{F} \otimes \mathbb{R}^P)$ be such that $g^*(x, p) = \mathsf{G}\mathsf{k}_{\mathsf{x},\mathsf{p}}$ for any $x \in X$ and $p \in P$. Let $G_1, \dots, G_P \in \mathcal{H} \otimes \mathcal{F}$ the operator such that $G_p \eta = G(\eta \otimes e_p)$ for any $p \in P$ and $\eta \in \mathcal{F}$. Then, there exists an operator $\mathsf{H} \in (\mathcal{H} \otimes \mathbb{R}^P) \otimes \mathcal{F}$, such that*

- $\mathsf{H}\bar{k}_x = h^*(x)$ *for all $x \in X$.*

- $\|G\|^2_{\mathrm{HS}(\mathcal{F} \otimes \mathbb{R}^P, \mathcal{H})} = \sum_{p \in P} \|\mathsf{G}_p\|^2_{\mathrm{HS}(\mathcal{F}, \mathcal{H})}.$

- $\|H\|^2_{\mathrm{HS}(\mathcal{F}, \mathcal{H} \otimes \mathbb{R}^P)} = \sum_{p \in P} \pi(p) \|\mathsf{G}_p\|^2_{\mathrm{HS}(\mathcal{F}, \mathcal{H})}.$

*Proof.* Note that since $e_p$ form a basis of $\mathbb{R}^P$, we can write $G = \sum_{p \in P} G_p \otimes e_p$ and therefore

$$\|G\|^2_{\mathrm{HS}(\mathcal{F} \otimes \mathbb{R}^P, \mathcal{H})} = \sum_{p \in P} \|\mathsf{G}_p\|^2_{\mathrm{HS}(\mathcal{F}, \mathcal{H})} \tag{241}$$

as required.

Now, by definition of $h^*$ and the relation with $g^*$, we have that

$$h^*(x) = (\sqrt{\pi(p)}\, g^*(x, p))_{p \in P} \tag{242}$$
$$= (\sqrt{\pi(p)}\, \mathsf{G}\mathsf{k}_{x,p})_{p \in P} \tag{243}$$
$$= \left(\sqrt{\pi(p)}\, \mathsf{G}(\bar{k}_x \otimes e_p)\right)_{p \in P} \tag{244}$$
$$= \left(\sqrt{\pi(p)}\mathsf{G}_p \bar{k}_x\right)_{p \in P} \tag{245}$$
$$= \mathsf{H}\bar{k}_x, \tag{246}$$

where we have denoted with $\mathsf{H} \in (\mathcal{H} \otimes \mathbb{R}^P) \otimes \mathcal{F}$, the operator from $\mathcal{F}$ to $\mathcal{H} \otimes \mathbb{R}^P$, such that for any $\eta \in \mathcal{F}$ we have $\mathsf{H} = \left(\sqrt{\pi(p)}\mathsf{G}_p \eta\right)_{p \in P}$. The required results follow directly from the construction of both $\mathsf{G}$ and $\mathsf{H}$ in terms of the $\mathsf{G}_p$ for $p \in P$. $\qquad\square$

We can therefore conclude the equivalence between the original SELF estimator with kernel $\bar{k}$ and the SELF estimator by parts considered in this work, with kernel $k$, under the assumption that $g^*$ (and equivalently $h^*$) belong to the corresponding RKHS.

**Theorem 33.** *The SELF estimator with kernel $\bar{k}$ has same rates as the SELF by parts with kernel $k$*

For simplicity, assume $\pi(p|x) = \frac{1}{|P|}$ for every $x \in X$ and $p \in P$. From (6) and the SELF assumption, we have

$$\triangle(z, y|x) = \frac{1}{|P|} \sum_{p \in P} \langle \psi(z_p, x_p, p), \varphi(y_p) \rangle_{\mathcal{H}}. \tag{247}$$

Denote $\bar{\psi} : Z \times X \to \mathcal{H} \otimes \mathbb{R}^P$ and $\bar{\varphi} : Y \to \mathcal{H} \otimes \mathbb{R}^P$ the maps such that

$$\bar{\psi}(z, x) = \left(\psi(z_p, x_p, p)\right)_{p \in P} \qquad \bar{\varphi}(y) = \left(\varphi(y_p)\right)_{p \in P} \tag{248}$$

which can be interpreted as the concatenation of the different $\psi$ and $\varphi$ for $p \in P$. Then we can rewrite $\triangle$ in terms of the canonical inner product of $\mathcal{H} \otimes \mathbb{R}^P$,

$$\triangle(z, y|x) = \frac{1}{|P|} \langle \bar{\psi}(z, x), \bar{\varphi}(y) \rangle_{\mathcal{H} \otimes \mathbb{R}^P}. \tag{249}$$

We can now apply the approach proposed in this work to the case of a problem *with one single part* (or equivalently apply the SELF approach in [8]). The target function of this problem is $h^* : X \to \mathcal{H} \otimes \mathbb{R}^P$ defined as

$$h*(x) = \frac{1}{|P|} \int \bar{\varphi}(y)\, d\rho(y|x) = \frac{1}{|P|} \left( \int \varphi(y_p)\, d\rho(y|x) \right)_{p \in P} = \frac{1}{|P|} (g^*(x,p))_{p \in P} \in \mathcal{H} \otimes \mathbb{R}^P \tag{250}$$

and is the concatenation of all functions $g^*(\cdot, p)$ for $p \in P$.

Now, let us consider a rkhs $\mathcal{F}$ of functions $h : X \to \mathbb{R}$ with associated kernel $k : X \times X \to \mathbb{R}$. Assume that $h^*$ belongs to the space of vector valued functions $\mathcal{F} \otimes (\mathcal{H} \otimes \mathbb{R}^P)$. In other words, there exists an Hilbert-Schmidt operator $H : \mathcal{F} \to \mathcal{H} \otimes \mathbb{R}^P$ such that $Hk_x = h^*(x)$ for any $p \in P$. Note that this is equivalent to require that the function $g^*$ belongs to the space $(\mathcal{F} \otimes \mathbb{R}^P) \otimes \mathcal{H}$, namely that there exists an Hilbert-Schmidt operator, such that $G : \mathcal{F} \otimes \mathbb{R}^P \to \mathcal{H}$, such that, $G(k_x \otimes e_p) = g^*(x,p)$ for any $x \in X$ and $p \in P$, with $e_p \in \mathbb{R}^P$ denoting the $p$-th element of the canonical basis of $\mathbb{R}^P$. In particular, note that, for any $\eta \in \mathcal{H}$, $p \in P$ and $x \in X$, we have

$$\langle Hk_x, \eta \otimes e_p \rangle_{\mathcal{H}} = \langle h^*(x), \eta \otimes e_p \rangle = \langle h^*(x)_p, \eta \rangle_{\mathcal{H}} = \langle g^*(x,p), \eta \rangle_{\mathcal{H}} = \frac{1}{|P|} \langle G(k_x \otimes e_p), \eta \rangle. \tag{251}$$

We conclude that $H = \frac{1}{|P|}G$ and $\|H\|_{\mathrm{HS}} = \frac{1}{\sqrt{|P|}}\|G\|_{\mathrm{HS}}$. In particular, note that since $G \in (\mathcal{F} \otimes \mathbb{R}^P) \otimes \mathcal{H}$, we have that for any $p \in P$, the function $g(\cdot, p) : X \to \mathcal{H}$ is such that $g(\cdot, p) \in \mathcal{F} \otimes \mathcal{H}$. Therefore we have

$$\|G\|_{\mathrm{HS}} = \sqrt{\sum_{p \in P} \|g^*(\cdot, p)\|_{\mathcal{F} \otimes \mathcal{H}}^2}. \tag{252}$$

Interestingly, if all the functions $g^*(\cdot, p)$ have same norm $\mathfrak{g} = \|g^*(\cdot, p)\|_{\mathcal{F} \otimes \mathcal{H}}$ in $\mathcal{F} \otimes \mathcal{H}$, we have

$$\|H\|_{\mathrm{HS}} = \frac{1}{|P|}\|G\|_{\mathrm{HS}} = \frac{1}{\sqrt{P}} \sqrt{\sum_{p \in P} \mathfrak{g}^2} = \mathfrak{g}. \tag{253}$$

## I.4  The best of both worlds

Here we formalize the comment in Remark 6, where we introduced the kernel $k_B = k_U + k_L$ that is sum of a bounded universal continuous kernel $k_U$ over $X \times P$ and a bounded restriction (or "local") kernel $k_L$, satisfying (15). In particular we show that $k_B$ is universal but at the same time allows to train a structured prediction estimator $\widehat{f}$ that is able to leverage the locality of the learning problem, when available. For simplicity, we assume the input space $X$ to be compact and the set of parts indices $P$ to be finite.

Let $\mathcal{F}_B, \mathcal{F}_U$ and $\mathcal{F}_L$ denote the RKHSs of respectively $k_B$, $k_U$ and $k_L$. According to [3], we know that $\mathcal{F}_B \supseteq \mathcal{F}_U \cup \mathcal{F}_L$ and moreover that for any $h \in \mathcal{F}_B$, the norm is such that

$$\|h\|_{\mathcal{F}_B}^2 = \min_{h = h_U + h_L} \|h_U\|_{\mathcal{F}_U}^2 + \|h_L\|_{\mathcal{F}_L}^2, \tag{254}$$

with $h_U \in \mathcal{F}_U, h_L \in \mathcal{F}_L$. We immediately see that $k_B$ is universal. Indeed, since $k_U$ is universal, $\mathcal{F}_U$ is dense in the space of continuous functions on $X$ and consequently also $\mathcal{F}_B \supseteq \mathcal{F}_U$ is.

The following result is analogous to Cor. 24 and shows that the kernel $k_B$ is not only universal but also equivalent to $k_L$ in capturing the locality of the learning problem.

**Lemma 34.** *Denote by $k = k_B = k_U + k_L$ the sum kernel, where $k_U$ and $k_L$ are the universal and restriction kernels on $X \times P$, with $k_L$ as in (15) in terms of respectively $\bar{k} : [X] \times [X] \to \mathbb{R}$ and $k_0 : X \times X \to \mathbb{R}$. Let $\bar{r} = \sup_{\chi \in [X]} \bar{k}(\chi, \chi)$ and $r_0 = \sup_{x \in X} k_0(x,x)$.*

*Let $\pi(p|x) = \frac{1}{|P|}$ for any $x \in X$ and $p \in P$. Denote with $\bar{C}_{pq}$ the constant defined in (16) associated to the restriction kernel $k_L$. Then, the constant $\mathsf{q}$ in (167) associated to $k_B$ can be factorized as*

$$\mathsf{q} = \frac{1}{|P|^2} \sum_{p,q \in P} \mathsf{C}_{p,q}, \quad with \quad \mathsf{C}_{p,q} \leqslant \bar{\mathsf{C}}_{p,q} + (4\bar{r} + r_0) r_0\, \delta_{p,q}. \tag{255}$$

*Proof.* The proof of the result above follows by noting that, since $\pi$ is uniform, by Lemma 23, for any $p, q \in P$, $\mathsf{C}_{p,q}$ is characterized by

$$\mathsf{C}_{p,q} = \mathbb{E}_{x,x'}\left[(\bar{k}(x_p, x_q) + k_0(x,x)\delta_{p,q})^2 - (\bar{k}(x_p, x'_q) + k_0(x,x')\delta_{p,q})^2\right] \tag{256}$$

$$= \bar{\mathsf{C}}_{p,q} + \mathbb{E}_{x,x'}\left[k_0(x,x)^2 - k_0(x,x')^2\right]\delta_{p,q} + \tag{257}$$

$$- 2\mathbb{E}_{x,x'}\left[\bar{k}(x_p,x_q)k_0(x,x) - \bar{k}(x_p,x'_q)k_0(x,x')\right]\delta_{p,q} \tag{258}$$

$$\leqslant \bar{\mathsf{C}}_{p,q} + \delta_{p,q}\sup_{x\in X}k_0(x,x)^2 + 4\delta_{p,q}\left[\sup_{\chi\in[X]}\bar{k}(\chi,\chi)\sup_{x\in X}k_0(x,x)\right] \tag{259}$$

$$\leqslant \bar{\mathsf{C}}_{p,q} + (4\bar{\mathsf{r}} + \mathsf{r}_0)\,\mathsf{r}_0\,\delta_{p,q} \tag{260}$$

as desired. Note that the first inequality follows from the fact that $\bar{k}$ and $k_0$ are positive definite symmetric kernels. $\qquad\square$

Interestingly, Lemma 34 shows that the proposed sum kernel inherits the ability of the restriction kernel to capture the within- and between-locality of the learning problem. Combining this with the learning rates of Thm. 5, we obtain a result analogous to that of Thm. 4.

**Theorem 35** (Learning Rates & Locality). *With the same notation of Lemma 34 let $k_U$ be a bounded continuouous universal kernel on $X$, $k_L$ be the restriction kernel based on the reproducing kernel $\bar{k}$ on $[X]$ and let $\bar{\mathcal{F}}$ be the RKHS associated to $\bar{k}$. Let $\widehat{f}$ be the structured prediction estimator of (7) learned with kernel $k = k_B = k_U + k_L$. Then*

- *$\widehat{f}$ is universally consistent,*

- *Under Assumptions 1 and 2 and $\pi(p|x) = \frac{1}{|P|}$ for $x \in X, p \in P$, let $\bar{g}^*$ be defined as in Lemma 3 and $\bar{g}^* \in \mathcal{H} \otimes \bar{\mathcal{F}}$. Denote by $\bar{\mathsf{g}}$ the norm $\bar{\mathsf{g}} = \|\bar{g}^*\|_{\mathcal{H}\otimes\bar{\mathcal{F}}}$. When $\lambda = (\mathsf{r}^2/m + \mathsf{q}/n)^{1/2}$, then*

$$\mathbb{E}\left[\mathcal{E}(\widehat{f}) - \mathcal{E}(f^*)\right] \leqslant 12\,\mathsf{c}_\triangle\,\bar{\mathsf{g}}\,\mathsf{r}^{1/2}\left(\frac{1}{m} + \frac{c_1}{|P|n} + \frac{\sum_{p\neq q}e^{-\gamma d(p,q)}}{|P|^2 n}\right)^{1/4}, \tag{261}$$

*where $\mathsf{r} = \mathsf{r}_0 + \bar{\mathsf{r}}$, with $\mathsf{r}_0, \bar{\mathsf{r}}$ defined as in Lemma 34 and $c_1 = 1 + (4\bar{\mathsf{r}} + \mathsf{r}_0)\,\mathsf{r}_0/\mathsf{r}^2$.*

*Proof.* Let $\mathcal{F}_B, \mathcal{F}_U$ and $\mathcal{F}_L$ denote the RKHSs of respectively $k_B$, $k_U$ and $k_L$.

First, as discussed at the beginning of this section, the kernel $k = k_B := k_U + k_L$ is universal, since $\mathcal{F}_U \subseteq \mathcal{F}_B$ (see [3]) and $\mathcal{F}_U$ is dense in the continuous functions on $X \times P$. Then we can directly apply Thm. 25 obtaining the unversal consistency for $\widehat{f}$.

Second, under Assumption 1, by Lemma 3, we have that there exists $\bar{g}^* : [X] \to \mathcal{H}$ such that $g^*$, defined as in (13), is characterized by $g^*(x,p) = \bar{g}^*(x_p)$. S ince we assume that $\bar{g}^* \in \mathcal{H} \otimes \bar{\mathcal{F}}$ and we are using a restriction kernel under between-locality, we can apply Lemma 22 (where we used $\bar{\mathcal{G}}$ to denote $\bar{\mathcal{F}}$ and $\mathcal{F}$ to denote $\mathcal{F}_L$ and $\bar{g}^* \in \mathcal{H} \otimes \bar{\mathcal{F}}$ is expressed more formally by Assumption 6), then $g^* \in \mathcal{H} \otimes \mathcal{F}_L$ and $\|g^*\|_{\mathcal{H}\otimes\mathcal{F}_L} = \|\bar{g}^*\|_{\mathcal{H}\otimes\bar{\mathcal{F}}}$. Now, according to (254) (see [3]), for any function $h \in \mathcal{F}_L$ we have

$$\|h\|_{\mathcal{F}_B} := \min\{\|h_U\|_{\mathcal{F}_U} + \|h_L\|_{\mathcal{F}_L} \mid h = h_U + h_L, h_U \in \mathcal{F}_U, h_L \in \mathcal{F}_L\} \leqslant \|h\|_{\mathcal{F}_L},$$

since $h$ can be always decomposed as $h = h_L + h_U$ with $h_L = h$ and $h_U = 0$, then $\|g^*\|_{\mathcal{H}\otimes\mathcal{F}_B} \leqslant \|g\|_{\mathcal{H}\otimes\mathcal{F}_L}$. So

$$\|g^*\|_{\mathcal{H}\otimes\mathcal{F}_B} \leqslant \|\bar{g}^*\|_{\mathcal{H}\otimes\bar{\mathcal{F}}}.$$

Now we are ready to apply Thm. 5, with $\lambda = \sqrt{\mathsf{r}^2/m + \mathsf{q}/n}$ obtaining

$$\mathbb{E}\left[\mathcal{E}(\widehat{f}) - \mathcal{E}(f^*)\right] \leqslant 12\,\mathsf{c}_\triangle\,\bar{\mathsf{g}}\left(\frac{r^2}{m} + \frac{q}{n}\right)^{1/4}. \tag{262}$$

Finally note that since $\pi(p|x) = \frac{1}{|P|}$ for $p \in P, x \in X$, we can apply Lemma 34

$$\frac{q}{n} = \frac{\mathsf{r}^2 c_1}{|P|n} + \frac{\mathsf{r}^2\sum_{p\neq q}e^{-\gamma d(p,q)}}{|P|^2 n},$$

obtaining the desired result. $\qquad\square$

---

**Algorithm 2** – LEARN $\widehat{f}$

---

**Input:** training set $(x_i, y_i)_{i=1}^n$, distributions $\pi(\cdot|x)$ and $\mu(\cdot|y, x, p)$, reproducing kernel $k$ on $X \times P$, hyperparameter $\lambda > 0$, auxiliary dataset size $m \in \mathbb{N}$.

GENERATE the auxiliary dataset $(w_j, x_{i_j}, p_j)_{j=1}^m$:
    Sample $i_j$ uniformly from $\{1, \ldots, n\}$
    Sample $p_j \sim \pi(\cdot|x_{i_j})$
    Sample $w_j \sim \mu(\cdot|y_{i_j}, x_{i_j}, p_j)$

LEARN the coefficients for the score function $\alpha$:
    $\mathbf{K} \in \mathbb{R}^{m \times m}$ with entries $\mathbf{K}_{jj'} = k\big((x_{i_j}, p_j), (x_{i_{j'}}, p_{j'})\big)$
    $\mathbf{A} = (\mathbf{K} + m\lambda I)^{-1}$

**Return:** $\alpha : X \times P \to \mathbb{R}^m$ such that $\alpha(x, p) = \mathbf{A}\, v(x, p)$ with $v(x, p) \in \mathbb{R}^m$ is the vector with entries $v(x, p)_j = k\big((x_{i_j}, p_j), (x, p)\big)$.

---

---

**Algorithm 3** – EVALUATING $\widehat{f}$

---

**Input:** input $x \in X$, distribution $\pi(\cdot|x)$, auxiliary dataset $(w_j, x_{i_j}, p_j)_{j=1}^m$, score functions $\alpha : X \times P \to \mathbb{R}$, number of iterations $T$, step sizes $\{\gamma_t\}_{t \in \mathbb{N}}$.

INITIALIZE: $z_0 = 0$

**For** $t = 1$ to $T$
    Sample $p \sim \pi(\cdot|x)$
    $A(x, p) = \sum_{j=1}^m |\alpha_j(x, p)|$
    Sample $j$ from $\{1, \ldots, m\}$ with $\mathbb{P}(j = k) = |\alpha_k(x, p)|/A(x, p)$
    $h_{j,p} = \text{sign}(\alpha_j(x, p))\, A(x, p)\, \ell(z, w_j|x, p)$
    Choose $u \in \partial h_{j,p}(\cdot|x)(z_{t-1})$
    $z_t = \text{proj}_Z(z_{t-1} - \gamma_t u)$

**Return**: $z_T$

---

The discussion above implies that under the locality assumptions, the rates in Thm. 35 are essentially equivalent to the ones of the estimator trained with only the restriction kernel in Thm. 4.

## J   Additional details on evaluating $\widehat{f}$

According to (7), evaluating $\widehat{f}$ on a test point $x \in X$ consists in solving an optimization problem over the output space $Z$. This is a standard procedure in structured prediction settings [29], where a corresponding optimization method is derived on a case-by-case basis depending on the loss and the space $Z$ ([29]). However, the specific form of the objective functional characterizing $\widehat{f}$ in our setting allows to devise a general stochastic meta-algorithm to solve such problem. We observe that (7) can be rewritten as

$$\widehat{f}(x) = \underset{z \in Z}{\text{argmin}}\; \mathbb{E}_{(j,p)}\, h_{j,p}(z|x) \tag{263}$$

where for any $p \in P$ and $j \in \{1, \ldots, m\}$ we have introduced the functions $h_{j,p} : Z \to \mathbb{R}$, such that

$$h_{j,p}(\cdot|x) = \big(\,\text{sign}(\,\alpha_j(x, p)\,)\, \mathsf{A}(x, p)\big)\, \ell(\cdot, w_j|x, p) \tag{264}$$

with $\mathsf{A}(x, p) = \sum_{j=1}^m |\alpha_j(x, p)|$. In the expectation above, the variable $p$ is sampled according to $\pi(\cdot|x)$ and $j$ is sampled from the set $\{1, \ldots, m\}$ with probability $\frac{|\alpha_j(x,p)|}{\mathsf{A}(x,p)}$. When the $h_{j,p}$ are (sub)differentiable, problems of the form of (11) can be addressed by stochastic gradient methods (SGM). In Alg. 3 in the supplementary material we provide an example of such strategy.

# K  Additional examples of Loss Functions by Parts

Several structured prediction settings are recovered within the setting considered in this work and the associated loss functions have the form of (6). Below recall some of the most relevant examples where the locality assumptions can be reasonaly expected to hold.

**Hamming.** A standard loss function used in structured prediction is the Hamming loss [10, 42, 11], which for any factorization by parts can be written as in (6) with $L_p(z_p, y_p|x_p) = \delta(z_p \neq y_p)$, the function equal to 0 if $z_p = y_p$ and 1 otherwise.

- **Computer Vision**. The Hamming loss is often used in computer vision [29, 45]. For instance, in image segmentation [41] the goal is to label each pixel $p$ of an input image $x$, as background ($y_p = 0$) or foreground ($y_p = 1$). Errors are measured as total number of mistakes $z_p \neq y_p$ over the total number of pixels.

- **Hierarchical Classification**. In classification settings with a hierarchy [44], errors are weighted according to the semantic distance between two classes (e.g. classifying the image of a "dog" as a "bus" is worse than classifying it as a "cat"). Assuming the hierarchy between classes to be represented as a tree, these loss functions can be written as the Hamming loss between the parts of a class $y = (y_{\text{root}}, \ldots, y_{\text{leaf}})$ seens as the collection of all the nodes in its hierarchy (e.g. "cat", "feline", "mammal", "animate object", "entity").

- **Planning**. In learning-to-plan applications [31], the goal is to predict a trajectory $z$ closest to a ground truth trajectory (typically provided by an expert). A trajectory is represented as a sequence of contiguous states $y = (y_{\text{start}}, \ldots, y_{\text{end}})$ and errors with respect to a predicted trajectory $z$ are measured in terms of the number of states that do not coincide, namely the hamming loss between the two sequences.

This loss has been extensively used in computer vision for applications such as pixel-wise classification [41] or image segmentation [1].

**Precision/Recall, F1 Score..** The precision/recall and F1 score are loss functions often adopted in natural language processing [43]. They are used to measure the similarity between two binary sequences. Given two binary sequences $z, y \in \{0, 1\}^k$ of length $k$, we have $\triangle(z, y) = \triangle(z^\top y, \|z\|^2, \|y\|^2)$. In particular, the precision correponds to $\triangle(z, y) = z^\top y / \|z\|^2$, the recall to $\triangle(z, y) = z^\top y / \|y\|^2$ and the F1 score to $\triangle(z, y) = z^\top y / (\|z\|^2 + \|y\|^2)$. These functions are in the form of (6) if taking $|P| = k$ and $i_Y(y, p) = (y_p, \|y\|)$, $i_Y(z, p) = (z_p, \|z\|)$. Note that the number of elements in $y$ and $z$ can vary depending on the cardinality $|x|$ of each input $x$, (see e.g. [43]). In this sense the $\triangle(z, y|x)$ is necessarily parametrized by $x$ and in particular the set $P$ is a set $P(x) = \{1, \ldots, |x|\}$.

**Multitask Learning.** Multitask learning settings have a natural decomposition into parts: the output and label spaces $Z$ and $Y$ are subset of $\mathbb{R}^T$, and $\triangle(z, y) = \frac{1}{T} \sum_{t=1}^{T} L(z_t, y_t)$, with $L$ any loss function commonly used in standard supervised learning problems (e.g. least-squares for regression, hinge or logistic for classification). In settings where $Z$ is not a linear space but a *constraint set*, our model recovers the non-linear multitask learning framework considered in [9].

**Learning sequences..** Let $X = A^k$, $Y = Z = B^k$ for two sets $A, B$ and $k \in \mathbb{N}$ a fixed length. We consider a set of structures $P \subseteq \mathbb{N}^2$ such that any pair $p = (s, l) \in P$ indicates the starting element and the length of a subsequence. In particular, we choose the set of parts $\mathcal{X} = \cup_{t=1}^{k} A^t$ and $\mathcal{Y} = \mathcal{Z} = \cup_{t=1}^{k} B^t$ with

$$x_p = (x^{(s)}, \ldots, x^{(s+l-1)}) \in \mathcal{X} \qquad \forall\, x \in X, \ \forall\, (s, l) \in P \qquad (265)$$

where we have denoted $x^{(s)}$ the $s$-th entry of the sequence $x \in X$. Analogously $y_p = (y^{(s)}, \ldots, y^{(s+l-1)})$ for $y \in Y$. Finally, we choose the loss $L_0$ to be the (normalized) edit distance between two strings of same length

$$L_0(z, y; x, (s, l)) = \frac{1}{l} \sum_{i=1}^{l} \mathbf{1}(z^{(i)} \neq y^{(i)}) \qquad (266)$$

where $\mathbf{1}(z^{(i)} \neq y^{(i)}) = 0$ if $z^{(i)} = y^{(i)}$ and 1 otherwise (clearly a generic loss function $h(z^{(i)} \neq y^{(i)})$ and weight $w_i$ can be used instead of $\mathbf{1}$ and $1/l$). Finally, we can choose the uniform distribution $\pi(p|x) = 1/|P|$ (but clearly also less symmetric weighting strategy can be adopted).

**Pixelwise classification on images.** Consider the problem of assigning each pixel of an image to one of $T$ separate classes. In this setting $X = \mathbb{R}^{d \times d}$ is the set of images (with fixed width and height equal to $d \in \mathbb{N}$) and $Y = Z = \mathbb{R}^{T \times d \times d}$ is the set of all possible ways to label an image. We choose the set of parts $\mathcal{X} = \cup_{w,h=1}^{d} \mathbb{R}^{w \times h}$ to be the set of all possible patches of $d \times d$ image and the set of structures to be a $P \subset \mathbb{N}^4$ such that for any image $x \in X$ and $p = (u, l, w, h) \in P$ the selectors $x_p \in \mathbb{R}^{w \times h}$ and $y_p, z_p \in \mathbb{R}^{T \times w \times h}$ correspond to the patch of the image $x$ or the labeling $y$ and $z$ with width $w$, height $h$ and upper-left corner at the pixel $(u, l)$.

We choose the loss $L_0$ to be a function comparing the class "statistics" in a given patch: e.g.

$$L_0(z_p, y_p; x_p, p) = \|\sigma(z_p) - \sigma(y_p)\|^2 \qquad \sigma(\zeta) = \frac{\sum_{i=1}^{\text{width}(\zeta)} \sum_{j=1}^{\text{height}(\zeta)} \zeta_{:,i,j}}{\text{width}(\zeta)\text{height}(\zeta)}. \tag{267}$$

Since it is more likely to have larger values for $L_0$ at higher scales (the object patch overlaps other classes), we choose a weighting $\pi(p|x)$ that is decreasing with respect to the size of the patch $p = (u, l, w, h)$. For instance we can choose $\pi(p|x) = \frac{\exp(-\gamma wh)}{\sum_{p'=(u',l',w',h') \in P} \exp(-\gamma w'h')}$, for $\gamma > 0$.

## K.1 Example: Locality on sequences

We comment here on the example in Example 1 proving the inequality (4). We assume Assumption 2 to hold for $P = \{1, \ldots, |P|\}$ with $d(p, q) = |p - q|$ and $\gamma > 0$. We have

$$\mathsf{s} = \frac{\mathsf{r}^2}{|P|} \sum_{p,q=1}^{|P|} e^{-\gamma|p-q|} \tag{268}$$

$$\leqslant \frac{\mathsf{r}^2}{|P|} \sum_{p=q=1}^{|P|} e^{-\gamma|p-q|} + 2\frac{\mathsf{r}^2}{|P|} \sum_{p=1}^{|P|-1} \sum_{q>p}^{|P|} e^{-\gamma|p-q|}. \tag{269}$$

Now, we introduce the change of variable $t = q - p$ to obtain

$$\mathsf{r}^2 + 2\frac{\mathsf{r}^2}{|P|} \sum_{p=1}^{|P|-1} \sum_{\substack{t=1 \\ q=p+t}}^{|P|-p} e^{-\gamma|p-q|} = \mathsf{r}^2 + 2\frac{\mathsf{r}^2}{|P|} \sum_{p=1}^{|P|-1} \sum_{t=1}^{|P|-p} e^{-\gamma t} \tag{270}$$

$$\leqslant \mathsf{r}^2 + 2\frac{\mathsf{r}^2}{|P|} \sum_{p=1}^{|P|-1} \sum_{t=1}^{|P|} e^{-\gamma t} \tag{271}$$

$$\leqslant \mathsf{r}^2 + 2\mathsf{r}^2 \sum_{t=1}^{|P|} e^{-\gamma t} \tag{272}$$

$$\leqslant 2\mathsf{r}^2 (\sum_{t=0}^{|P|} e^{-\gamma t}). \tag{273}$$

We can upper bound $\sum_{t=1}^{|P|} e^{-\gamma t} = \sum_{t=1}^{|P|} (e^{-\gamma})^t$ with the geometric series $\sum_{t=1}^{+\infty} e^{-\gamma t}$. Since $\gamma > 0$ we conclude that such series is upper bounded by $(1 - e^{-\gamma})^{-1}$, concluding

$$\mathsf{s} \leqslant 2\mathsf{r}^2 (1 - e^{-\gamma})^{-1}, \tag{274}$$

as desired.