[Reviews · NeurIPS 2019]

Reviewer 1



- The authors propose a way to perform structured output prediction when the dependence between 'parts' are localized. The model is learned by breaking the structure into parts and performing kernel ridge regression on the parts. They show elaborate convergence rate analysis in the estimation. The theoretical analysis is the strong part of this paper. - One of my concern for this paper is whether the locality assumption made in this paper is valid in general. In a lot of computer vision and NLP applications the latest research is about capturing long range dependencies. The correlation in Figure 1 is highly concentrated at the central patch because it's the average of many different images, but on individual images the correlation patten can be very different. But the learning method does not capture these dependencies by breaking the learning problem as part-based regression. It would be useful to test this assumption on more real-world applications. - One difficulty with doing this learning by parts is inference, i.e., how to put the parts back together to make one single prediction during test time. The authors suggest using stochastic gradient in the evaluation of (7). But given the non-differentiability of most structured prediction problem wrt to the output Z, I am not sure this general scheme would perform well. Also, Algorithm 3 given in the Appendix involves projection onto the set of feasible structures Z, which can be very difficult. - Another concern for this paper is the range of applicability of this model. The method is evaluated on a single fingerprint direction prediction problem, and compared against several kernel ridge regression based methods. It would be more interesting to test on more traditional problems such as sequence learning, segmentation with MRF, etc, to see if the locality assumption used in the new method actually leads to any improvements. Given this is described as a general structured prediction method, more than one example application should be included to illustrate the model. For example, the authors can quite easily implement Example 2 suggested in their paper. - Given the proposed method is quite new compared to traditional structured prediction approaches, I believe more examples and experiments need to be given to make sure the model is not a just method with nice theoretical properties but limited real-world applications.

Reviewer 2



Strengths: The paper is very clear and the small parts remaining dark (such as what the authors refer to when dealing with unclearly defined parts or losses decomposable by part in general) are studied in detail in the supplementary. This works is a natural follow up to the previous work by Ciliberto et al which introduced a learning bound similar to Theorem 4 in a more general setting. Showing how to take advantage of the specificities of the data and characterising this in terms of improvement on statistical guarantees is very appreciated in this domain. Weaknesses: Despite the shown results and the details added in the appendix K, I think that the experimental part remains the weak part of this paper. The results displayed are convincing but I am disappointed that the authors did not tried their approach on more popular problems mentioned in the supplementary such as hierarchical classification. Even if this could be improved (in order to be at the level of the theoretical treatment), the proposed content is already solid and does not change my decision concerning the quality of this work. Remark: - The use of the sequence example at different step of the paper is really useful, however I'm a bit surprised that you mention in Example 2 a 'common' practice in the context of CRF corresponding to using as a scoring loss the Hamming distance over entire parts of the sequence. I've never seen this type of approach and am only aware of works reporting the hamming loss defined node wise. It would be great if you could point out some references there. - After reading the paper a few times, I still think that the notation $\Delta(z,y|x)$ is a bit strange and I would have preferred something of the form $\Delta(z,y)$ since in practice the losses you mention never takes into account the input data and $z$ is already a function of $x$. Maybe this is only personal taste and will be contradicted by the other reviewers. Minor remarks : missing brackets [ ] in theorem 4.

Reviewer 3



Originality ------------ The authors propose a new setting for structured prediction, and relates it well to the state of the art. By developing their new framework, the authors are also able to explain the efficiency of some existing structured prediction method (restriction kernel). Quality --------- Although I didn't go through all the proofs in the appendix, the paper looks technically sound. All the claims presented are proven in detail. The authors provide a very complete piece of work, and the appendix extend even more the proposed framework. Clarity -------- The paper is clearly written and organized. The authors provide all the elements to understand the paper, and it should be enough to reproduce the results. Figure 3, the Left plot is very difficult to read in black and white. Significance ---------------. The presented results are significant as it address a difficult task by exploiting structure in the data, and improve over existing methods. The proposed algorithm is proven to be consistent and the authors provide and study generalization rates, justifying the usefulness of the method. I have read the author response and my opinion remains the same.

[Author Response · NeurIPS 2019]

We kindly thank the reviewers for their feedback. All reviewers expressed their interest in more experimental evaluation.
We report an additional experiment on sequence prediction and then address each reviewer's questions individually.

**Sequence Prediction.** We considered the *TIMIT Acoustic-Phonetic Continuous Speech Corpus* (`https://catalog.`
`ldc.upenn.edu/LDC93S1`), containing recordings of $\sim 6300$ recorded sentences (630 English speakers, each reading
$\sim 10$ sentences). We preprocessed the dataset following a standard practice in the speech community, taking 10ms
frames (dropping the glottal stop 'q' labeled frames in the core test set) and mapping them as 40-dimensional vectors [
D. Povey et al., *"The kaldi speech recognition toolkit"*, 2011]. The speech recognition problem is stated in terms of
sequence prediction, i.e. each sentence, represented as a list of 40-dimensional vectors to be mapped in a same-length
list of phonemes (131 phonemes per each 10ms frame). We considered:

1. A baseline performing independent classification of sequence elements using Kernel SVM with Gaussian Kernel.
The regularization parameter $C$ and the bandwidth of the kernel have been selected via hold-out cross validation in
the logarithmic range $[10^{-3}, 10^3]$ and $[10^{-9}, 10^3]$.

2. Structural SVM for sequence tagging based on Hidden Markov Models (SVM-HMM) and exact Viterbi algorithm
for inference (see below). Implementation from `https://www.cs.cornell.edu/people/tj/svm_light/svm_`
`hmm.html` using Gaussian kernel. The regularization parameter $C$ and the bandwidth of the kernel have been
selected via hold-out cross validation in the logarithmic range $[10^{-3}, 10^3]$ and $[10^{-9}, 10^3]$.

3. The proposed *localized structured prediction algorithm*. In particular we defined parts on input as subsequences of
11 frames (110ms) corresponding to vectors of 440 elements, while the corresponding parts of the output are the
couple of phonemes associated to the central and the next frame of the input.

*Inference.* For both SVM-HMM and our method, the inference is solved with the classic Viterbi algorithm over a table
of dimension $L|x|$ where $L$ is the number of labels per element ($L = 131$ phonemes) and $|x|$ is the lenght of the input
sequence ($|x| \sim 200$ on average). When output parts are couples of phonemes this entiails $O(|x|L^2)$ computations.

23
*Results.* The Table reports the test performance (Hamming loss)
of the three methods over 5 trials (dataset is reshuffled and
randomly split in 3300 sentences for training and the rest for
test). Leveraging the parts structure appears beneficial, with
our method consistently outperforming both the baseline and
the traditional structured prediction competitor.

| Sequence prediction on TIMIT | Hamming |
|---|---|
| Independent SVM | $0.31 \pm 0.015$ |
| Struct SVM | $0.29 \pm 0.031$ |
| Localized Struct. Pred | **$0.26 \pm 0.012$** |

**R1** * *Long Range Dependecies.* Our assumption does not limit long-range dependencies in the data as they do not
imply that two parts to have identical (or very similar) appearance with respect to the similarity metric used. However,
it is also true that while such dependencies could be leveraged to improve performance, our approach is not designed to
capture them at training time (but only during inference). As future work we will explore the question of reformulating
the part-based regression as a multi-task learning (MTL) problem, with each task corresponding to a part. Then,
by leveraging ideas from the MTL literature we could impose (when known a-priori) or learn (when unknown) the
multi-scale relations/dependencies between tasks/parts during the training phase. Algorithmically this extension would
require a small modification of the current approach. We will add a comment on this when discussing future directions.
* *Inference.* The inference step in our setting is formulated as an optimization problem over the output space $Z$ as most
traditional structured prediction approaches (see Remark 2 in our paper). Hence, our inference step is as difficult as
previous methods, from a computational viewpoint (while also providing strong theoretical guarantees on the prediction).
For instance, in the experiments on TIMIT in this document, both our method and SVM-HMM use the viterbi algorithm
for inference. Algorithm 3 offers an additional benefit of our formulation when first order optimization on $Z$ is possible.

**R2** * *Example 2.* We agree with R2 to improve exposition of Example 2. In particular we will clarify the difference
between the loss function used to evaluate the error and the score function at inference time. While these are two
separate concepts in the context of CRFs, they essentially coincide in our Localized Structured Prediction framework
(score is a linear combination of losses). The conclusion of Example 2 highlights the connection between CRFs and
our framework by observing that the score functions have essentially the same structure. Hence they lead to the same
inference problem. *[Sutton, C and McCallum, A. "An introduction to conditional random fields." 2012].*
* *Notation* We agree with R2: will use $\Delta(z, y)$ in the main text and $\Delta(z, y|x)$ only in the appendix. This notation was
originally introduced to account for settings where the parts might depen on the input (e.g. images with differnt sizes).

**R3** **Attention Models.* Loosely speaking, an attention model could be formulated within our framwork by considering
a parametrization of the possible part structures: in this sense, the attention would consist in the process of selecting
iteratively the parts that are more relevant to the task and adapt them depending on individual inputs. This model
however would require to perform an optimization over the parts parametrization, making both learning and inference
significantly more challegning. Investigating this question could lead to interesting future work, we thank the reviewer
and we will add a comment to the paper. **Code.* Our code is in python (+ pytorch) and will be made available in Github.

[Meta-Review · NeurIPS 2019]

The authors propose a general theoretical framework for structured prediction that deals with cases where the data exhibits a local structure, so that the inputs and outputs can be decomposed into parts. The paper analyses the new framework theoretically and empirically. The reviewers deemed the theoretical contributions to be of original and of a high quality. The author response addressed the perceived weaknesses, in particular in the empirical evaluation, in a satisfcatory way.